# *map3k1* is required for spatial restriction of progenitor differentiation in planarians

Bryanna Isela-Inez Canales[1,2], Hunter O King[1,3], Peter W Reddien[1,2,4]*

[1]Whitehead Institute for Biomedical Research, Cambridge, United States; [2]Department of Biology, Massachusetts Institute of Technology, Cambridge, United States; [3]Department of Brain and Cognitive Sciences, Massachusetts Institute of Technology, Cambridge, United States; [4]Howard Hughes Medical Institute, Massachusetts Institute of Technology, Cambridge, United States

## eLife Assessment

This **important** study examines the role of map3k1, a MAP3K family member that has both kinase and ubiquitin ligase domains, in the differentiation of progenitors in the flatworm Planaria. The **convincing** analyses demonstrate that map3k1 acts within progenitors to restrict their premature differentiation and to prevent formation of teratomas. This work would be of interest to researchers in the fields of regeneration, developmental biology, and aging.

*For correspondence:
reddien@wi.mit.edu

**Competing interest:** The authors declare that no competing interests exist.

**Abstract** Planarian regeneration and tissue turnover involve fate specification in pluripotent stem cells called neoblasts. Neoblasts select fates through the expression of fate-specific transcription factors, generating specialized neoblasts. Specialized neoblasts are spatially intermingled and can be dispersed broadly, frequently being present far from their target tissue. The post-mitotic progeny of neoblasts, serving as progenitors, can migrate and differentiate into mature cell types. Pattern formation is thus strongly influenced by the migratory assortment and differentiation of fate-specified progenitors in precise locations, which we refer to as progenitor targeting. This central step of pattern maintenance and formation, however, is poorly understood. Here, we describe a requirement for the conserved *map3k1* gene in targeting, restricting post-mitotic progenitor differentiation to precise locations. RNAi of *map3k1* causes ectopic differentiation of eye progenitors along their migratory path, resulting in dispersed, ectopic eye cells and eyes. Other neural tissues similarly display ectopic posterior differentiation, and ectopic pharynx cells emerge dispersed laterally and anteriorly in *map3k1* RNAi animals. Ectopic differentiated cells are also found within the incorrect organs after *map3k1* RNAi, and ultimately, teratomas form. These findings implicate *map3k1* signaling in controlling the positional regulation of progenitor behavior – restricting progenitor differentiation to targeted locations in response to external cues in the local tissue environment.

## Introduction

Planarians are capable of regenerating any missing body part through the action of pluripotent stem cells called neoblasts (***Reddien, 2018***). Neoblasts maintain all cell types of the adult body through a process of constitutive cell turnover. Because planarians have >125 different adult cell types, these stem cells must be capable of choosing among a large array of possible cell fates (***Fincher et al., 2018***; ***Plass et al., 2018***; ***Zeng et al., 2018***; ***King et al., 2024***). Fate specification can occur in neoblasts through the activation of transcription factors called fate-specific transcription factors (FSTFs), producing specialized neoblasts (***Reddien, 2022***). Specialized neoblasts divide and can produce daughter cells that act as migratory precursors (post-mitotic progenitors) for differentiated

cell types (*Eisenhoffer et al., 2008*; *Wenemoser and Reddien, 2010*; *Guedelhoefer and Sánchez Alvarado, 2012*; *van Wolfswinkel et al., 2014*; *Abnave et al., 2017*; *Reddien, 2022*).

Fate choice in neoblasts can be regulated by position. For instance, eye-specialized neoblasts are formed in roughly the anterior third of the animal (*Lapan and Reddien, 2011*; *Lapan and Reddien, 2012*). However, this spatial regulation of stem cell fate specification is coarse when compared to the precise positions of differentiated cells associated with complex planarian tissue architecture (*Lapan and Reddien, 2011*; *Lapan and Reddien, 2012*; *Adler et al., 2014*; *Scimone et al., 2014a*; *van Wolfswinkel et al., 2014*; *Park et al., 2023*). Because specialized neoblasts are produced in broad regions, they are often found far from their target tissue. As a result, individual specialized neoblasts are frequently found closer to other differentiated cell types different than their target tissue (*Park et al., 2023*). Furthermore, neoblasts are spatially specified in a highly intermingled manner, in hetero-geneous neoblast neighborhoods (*Park et al., 2023*). For instance, a muscle-specialized neoblast could have a neural, intestinal, epidermal, protonephridial, or other specialized neoblast type as its nearest neoblast neighbor (*Park et al., 2023*). These observations suggest that the regulation of differentiation programs in post-mitotic migratory progenitors is a crucial aspect to patterning and tissue maintenance. Neoblasts, themselves, are not highly migratory under homeostatic conditions; however, the post-mitotic progenitor cells that they produce serve as precursors and can remain in an immature and sometimes migratory state for days until reaching their target tissue (*Saló and Baguñà, 1985*; *Eisenhoffer et al., 2008*; *Wenemoser and Reddien, 2010*; *Wagner et al., 2011*; *Guedel-hoefer and Sánchez Alvarado, 2012*; *van Wolfswinkel et al., 2014*; *Abnave et al., 2017*; *Park et al., 2023*). These findings generate a model for pattern maintenance during tissue turnover and formation in planarian regeneration in which specialized neoblasts generate intermingled post-mitotic progenitor classes that migrate to precise locations for differentiation (*Park et al., 2023*).

From a messy state of progenitor formation, order must arise. We hypothesized that terminal differentiation of post-mitotic progenitors is regulated to occur at precise positions to prevent disor-dered differentiation along migratory trails. This understudied mechanism could, in principle, be a major element of pattern formation from migratory progenitors in regenerative contexts. Under-standing how migratory progenitors know when and where to terminally differentiate into a mature stationary cell is a central problem for understanding how systems of migratory progenitors generate and maintain pattern. The regenerative biology of planarians presents the opportunity to uncover mechanisms underlying migratory progenitor targeting and differentiation regulation, which could apply to numerous developmental and regenerative contexts across the animal kingdom.

## Results

### *map3k1* RNAi results in the ectopic posterior differentiation of eye cells along the AP axis

We sought genes involved in processes that regulate where fate-specified progenitors differentiate through RNAi studies. We found a gene (*map3k1*, dd_5198) encoding a MAP3K1-like signaling protein for which RNAi resulted in a striking phenotype involving dispersed ectopic eyes and single eye cells (*Figure 1A*). The unique nature of this patterning phenotype raised the possibility that *map3k1* has some role in regulating progenitor differentiation during targeting. Planarian *map3k1* encodes a member of the MAP3K signaling protein family. Aside from a characteristic kinase domain, eukaryotic MAP3K1 proteins, including the planarian ortholog, possess unique accessory domains compared to other MAP3Ks; these include a PHD-like RING finger, a SWIM-type RING finger, and a TOG-like domain. These domains have been implicated in non-canonical MAP kinase signaling cascades that enable MAP3K1 to act as a ubiquitin ligase and a scaffold protein (*Lu et al., 2002*; *Xia et al., 2007*; *Figure 1—figure supplement 1A*).

The bilaterally symmetric anterior location of planarian eyes represents the normal targeting location of eye progenitors (*Lapan and Reddien, 2011*; *Lapan and Reddien, 2012*). RNAi of *map3k1* resulted in the gradual emergence of ectopic eyes posterior to the normal eye location with variability in their anterior–posterior (AP) and medial–lateral (ML) positions (*Figure 1A*). A similar *map3k1* RNAi phenotype was independently described in *Lo and Petersen, 2025*. Fluores-cent in situ hybridization (FISH) experiments showed that both optic cup (OC) cells and photore-ceptor neurons (PRNs) were present ectopically in clusters of cells and as individual, isolated cells

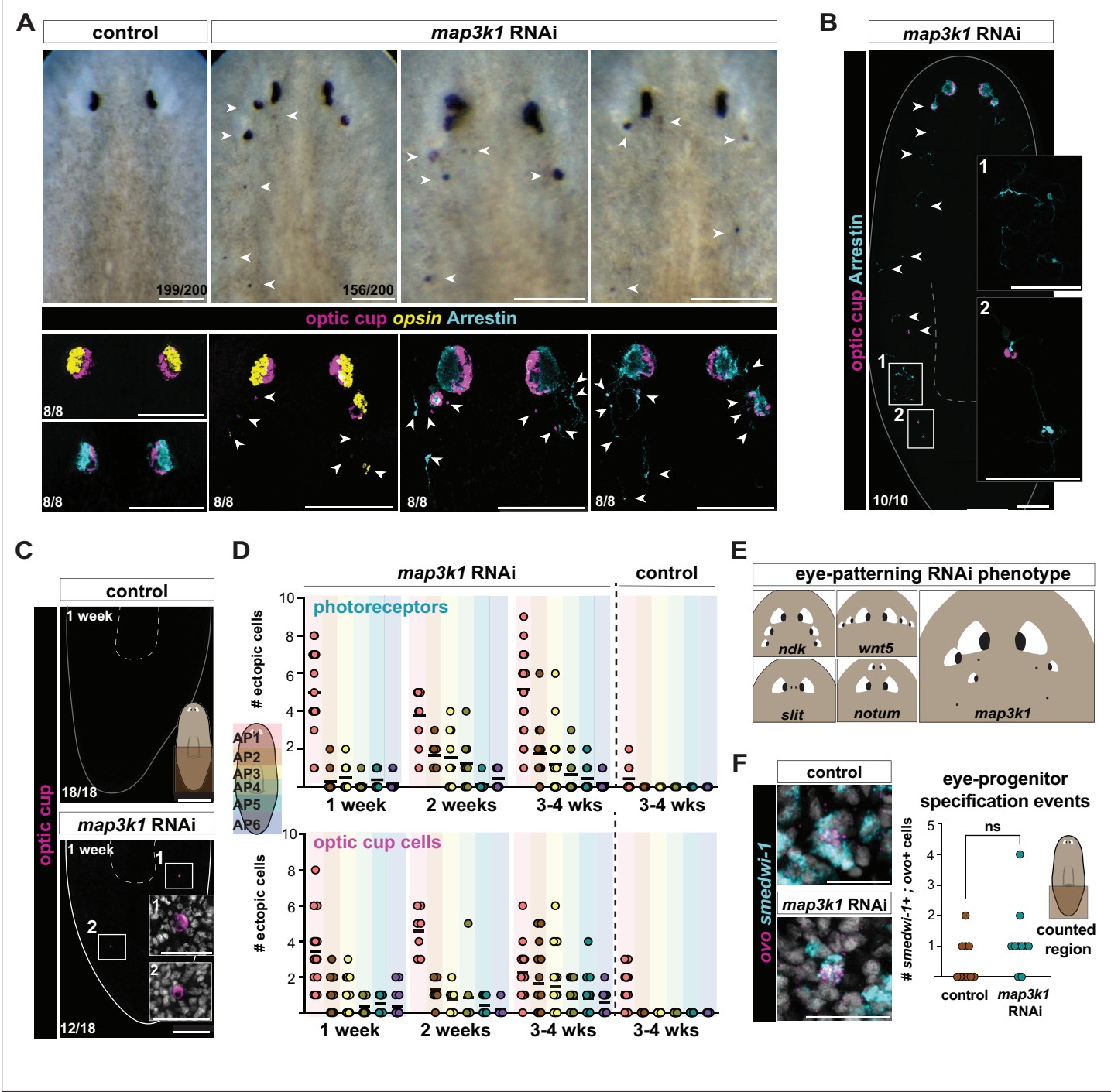

**Figure 1.** *map3k1* RNAi results in ectopic eyes and ectopic isolated eye cells. (**A**) Top row: live image of a control RNAi animal followed by three examples of *map3k1* RNAi animals with disorganized eyes (white arrows) (*n* = 156/200>1 ectopic eye, after 4 weeks of RNAi). Bottom row: control RNAi fluorescent in situ hybridization (FISH) images followed by three *map3k1* RNAi examples visualizing OC cells (RNA probe pool to *catalase/tyrosinase/glut3*); photoreceptor neurons (PRNs) are visualized with either an RNA probe to *opsin* (PRN cell bodies) or an anti-Arrestin antibody (PRN cell bodies and projections). The left and middle *map3k1* RNAi FISH examples show single ectopic PRNs (*opsin*, left; anti-Arrestin, middle) and OC cells (*catalase/tyrosinase/glut3*) scattered below and around the eyes after 3 weeks of RNAi. The far FISH example shows ectopic OC cells and PRNs (anti-Arrestin) after 6 weeks of RNAi. Dorsal up. Scale bars, 100 μm. (**B**) FISH showing ectopic PRNs (anti-Arrestin) and OC cells (*catalase/tyrosinase/glut3*) in the trunk and tail region of a *map3k1* RNAi animal after 5 weeks of RNAi (*n* = 10/10 with at least one cell in the trunk) (see panel A for control RNAi). Scale bar, 100 μm. Magnified panels 1 and 2, scale bars, 50 μm. White arrows point to all ectopic cells. (**C**) FISH showing the tail regions of control and *map3k1* RNAi animals; single OC (*catalase/tyrosinase/glut3*) cells are observed in the tail after 1 week of *map3k1* RNAi (*n* = 12/18 with at least one cell in the tail). Scale

*Figure 1 continued on next page*

*Figure 1 continued*

bar, 200 μm. (**D**) Top graph depicts the number of ectopic PRNs (1 week: *n* = 15; 2 weeks: *n* = 9; 3–4 weeks: *n* = 22). Control shown at 3–4 weeks (*n* = 11). Bottom graph depicts the number of ectopic OC cells (1 week: *n* = 16; 2 weeks: *n* = 7; 3–4 weeks: *n* = 17) per *map3k1* RNAi animal along the AP axis over time. Control shown at 3–4 weeks (*n* = 18). *map3k1* RNAi resulted in higher ectopic cell numbers along the AP axis (p < 0.0001; Poisson generalized linear mixed model) compared to the control condition for both PRNs and OC cells at 3–4 weeks. (**E**) Schematic comparing previously identified eye-patterning RNAi phenotypes (after *ndk, wnt5, slit,* and *notum* RNAi) with the *map3k1* RNAi phenotype. (**F**) FISH examples of eye-specialized neoblasts (*ovo*+; *smedwi-1*+ cells) in tails of control and *map3k1* RNAi animals at 3 weeks of RNAi. Scale bar, 20 μm. The right graph shows no significant difference (p = 0.181; permutation test, 10,000 permutations; two-tailed) in the frequency of eye-specialized neoblasts in the tails of control (*n* = 10) and *map3k1* RNAi (*n* = 10) animals. All images, dorsal up. Bottom left numbers indicate the number of animals exhibiting the shown phenotype out of the total number of animals observed.

The online version of this article includes the following figure supplement(s) for figure 1:

**Figure supplement 1.** *map3k1* RNAi results in differentiated eye cells throughout the AP axis, and no overt change in eye-progenitor distribution.

(*Figure 1A*). Isolated, individual PRN or OC cell differentiation is rarely observed in the wild-type state. Although ectopic eye cells appeared most frequently in the normal zone of eye-progenitor specification (the head), they also appeared throughout the trunk and tail – far from the canonical eye-progenitor specification zone (*Figure 1B, C*; *Figure 1—figure supplement 1B*). Isolated eye cells were observed in animal tails within 1–2 weeks of the first RNAi feeding when no, or very few other ectopic eye cells appeared between the head and tail (*Figure 1C, D*; *Figure 1—figure supplement 1B*). The far posterior ectopic eye cells that appeared in *map3k1* RNAi animals were sparser than those in the head and usually appeared as singletons rather than aggregates (*Figure 1C*; *Figure 1—figure supplement 1B*).

The precise location in the planarian body where dispersed eye progenitors migrate toward and target for differentiation is referred to as the target zone (TZ) (*Atabay et al., 2018*). A variety of genes can be inhibited to result in eye formation in ectopic positions along the AP and ML axes and are presumed to impact the location of the TZ: for instance, posterior TZ shifting following RNAi of *nou darake (ndk)* (*Cebrià et al., 2002*) and *wntA* (*Kobayashi et al., 2007*); anterior shifting following RNAi of *notum* (*Hill and Petersen, 2015*) and *nr4A* (*Li et al., 2019*); lateral shifting following RNAi of *wnt5*; and medial shifting following RNAi of *slit* (*Oderberg et al., 2017*; *Atabay et al., 2018*). However, in all of these cases, organized ectopic eyes appear along particular AP or ML trajectories (*Figure 1E*, summary cartoon), consistent with alteration of the eye-progenitor TZ along a single axis, but still with differentiation constrained to occur in a particular position on the orthogonal axis. By contrast, *map3k1* RNAi ectopic eyes and eye cells were more disorganized, often being ectopic in both AP and ML position (*Figure 1A*). This distinct phenotype raised the possibility that *map3k1* is required for some patterning process not previously disrupted by the inhibition of other patterning genes.

Patterning in planarians prominently involves constitutive and regional expression of genes constituting positional information, referred to as position control genes (PCGs) (*Reddien, 2018*). PCGs are predominantly expressed in planarian muscle (*Witchley et al., 2013*). *ndk, notum, nr4A, wnt5,* and *slit* are regulated in their spatial expression and are components of this PCG patterning system. Therefore, these genes likely influence progenitor-extrinsic cues that guide progenitors to particular locations. By contrast, *map3k1* was not overtly expressed in a spatially restricted manner; instead, it was expressed broadly across tissues (*Figure 1—figure supplement 1C*).

The appearance of eye cells in the tail of *map3k1* RNAi animals suggests that some eye-progenitor specification likely occurred in the tails of these animals. Eye-specialized neoblasts can be recognized by the co-expression of the eye-specific FSTF *ovo* (*Lapan and Reddien, 2012*) and the neoblast marker *smedwi-1* (*Reddien et al., 2005*). *ovo*+; *smedwi-1*+ neoblasts were observed in the midbody and tail of *map3k1* RNAi animals (*Figure 1F*; *Figure 1—figure supplement 1D, E*). However, the frequency of observed *ovo*+ neoblasts in the tail was very low, and a similar low frequency of *ovo*+ neoblasts was observed in control animal tails (*Figure 1F*; *Figure 1—figure supplement 1E*). There was therefore no overt change to the spatial pattern of eye-specialized neoblasts after *map3k1* RNAi. This raises the possibility that there exists a natural low frequency of sporadic eye-neoblast specification events outside of the predominant, anterior eye-progenitor specification zone and that these cells can ectopically differentiate into mature eye cells in the posterior when *map3k1* is inhibited.

## *map3k1* prevents the posteriorization of some but not all anterior cell types

To assess whether the ectopic pattern of differentiated cells in *map3k1* RNAi animals was specific to the eye, we visualized other differentiated tissue patterns. Laterally projecting neural branches in the central nervous system are normally restricted to the anterior planarian brain (*Hyman, 1951*). After *map3k1* RNAi, however, ectopic brain branches labeled by the neuronal markers *choline acetyltransferase* (*chat*) and *prohormone convertase-2* (*pc2*) emerged from the two ventral nerve cords, appearing most frequently in the anterior and mid-body of the animal (*Figure 2A*; *Figure 2—figure supplement 1A*). In some instances, branches were observed in the tail (*Figure 2—figure supplement 1B*). Ectopic eyes sent axons (labeled with an anti-Arrestin antibody) that traveled along these ectopic neural branches and main nerve cord tracts (*Figure 2A*; *Figure 2—figure supplement 1A, B*). Ectopic brain branches that emerged from ventral nerve cords contained *GluR*+ (dd_16476) neurons, confirming that they are at least partly composed of normally brain-branch-restricted neurons (*Figure 2B*). *GluR*+ neurons are similarly found in ectopic brain branches observed after RNAi of *ndk* (*Cebrià et al., 2002*).

The anterior-restricted population of dd_17258+ neurons also displayed ectopic differentiation across the AP axis of *map3k1* RNAi animals, extending to the tail tip (*Figure 2C*; *Figure 2—figure supplement 1C, D*). The correct number of dd_17258+ neurons, however, remained in the normal AP location (AP_1) (*Figure 2D*). By contrast, some other anterior neural populations, *cintillo*+ and *glutamic acid decarboxylase*+ (*gad*+) neurons, remained unaffected in *map3k1* RNAi animals (*Figure 2C*; *Figure 2—figure supplement 1C*). These findings suggest that *map3k1* is required for the normal AP restriction of a subset of neural cell types during tissue turnover.

To determine if tissue posteriorization in *map3k1* RNAi occurred for non-neuronal cell types other than OC cells, we assayed gland cell populations (dd_9223, dd_7131+, and dd_8476+) that normally reside in the head, with a small fraction extending posteriorly to the pharynx and rarely to the tail. *map3k1* RNAi animals showed posteriorization of gland cell distributions for the primarily anterior dd_7131+ and dd_8476+ populations (*Figure 2E*). As was the case with neuron types, not all gland cell types were affected by *map3k1* RNAi. dd_9223+ cells, the most anteriorly restricted of the three gland cell populations assessed, did not change in distribution (*Figure 2E*). These data indicate that *map3k1* broadly affects the spatial distribution of various differentiated cells during tissue maintenance, but that this role is restricted to a subset of cell types.

## *map3k1* inhibition causes ectopic anterior differentiation of pharynx progenitors

Another regional tissue that is maintained through turnover from regional progenitors is the planarian pharynx. Neoblasts that produce pharynx progenitors are broadly located in the trunk region of the animal and express *FoxA* (*Adler et al., 2014*; *Scimone et al., 2014a*). Pharynx progenitors enter the pharynx through a connection to the body at the anterior end of the organ, and this requires that progenitors are capable of moving in multiple directions as a response to extrinsic cues. After 3 weeks of *map3k1* RNAi, ectopic single *vitrin*+ cells were present around the pharynx and at the anterior end of the typical *FoxA*+ zone (*n* = 20/20 animals), even reaching the head region in some cases (*Figure 3A*). These ectopic cells occupied variable AP and ML locations between the original pharynx and the brain (*Figure 3A*, *Figure 3—figure supplement 1A*). This variable placement of ectopic pharyngeal cells on the ML axis was reminiscent of the patterning defect observed for eyes following *map3k1* RNAi, described above. Scattered ectopic foci of cells expressing *mhc-1* (a gene expressed in pharyngeal muscle) were present in the anterior half of the animal, and frequently near ectopic *vitrin*+ pharyngeal cells after 3 weeks of RNAi (*Figure 3B*). Clusters of pharyngeal cell types were present between the cephalic ganglia and in the pre-pharyngeal region by 6–8 weeks of *map3k1* RNAi (*Figure 3—figure supplement 1B*).

The planarian mouth is an epidermal opening at the posterior end of the pharynx. Following 3 weeks of *map3k1* RNAi, ectopic mouth cells marked by NB.22.1e appeared anterior to the typical mouth location as an anterior streak stemming from the original mouth. Rarely (*n* = 3/18 animals), an ectopic focus of NB.22.1e+ mouth cells was observed in the tail, posterior to the pharynx (*Figure 3—figure supplement 1C*). After 8 weeks of RNAi, ectopic scattered mouth cells were present lateral to the midline (*Figure 3C*). Because differentiated pharynx cells were observed outside of the canonical pharynx progenitor-specification zone, we considered the possibility that pharynx progenitor

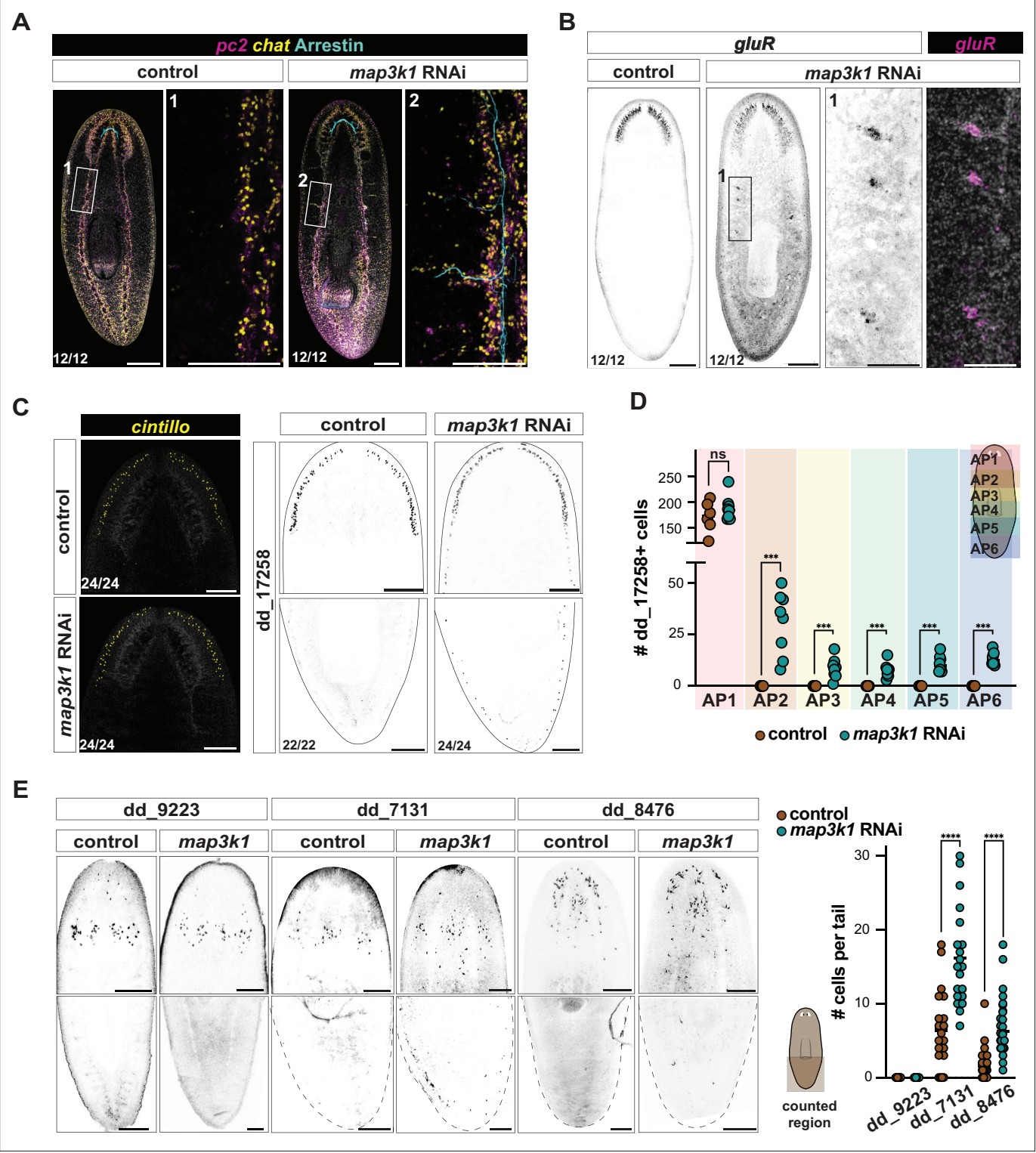

**Figure 2.** *map3k1* RNAi results in ectopic posterior differentiation of some neurons and gland cells. (**A**) *map3k1* RNAi animals exhibit posterior, ectopic brain branches that project dorsolaterally from ventral nerve cords (5 weeks of RNAi). These branches contain *chat*⁺ and *pc2*⁺ neurons. Photoreceptor axons (visualized with an anti-Arrestin antibody) extend along the ventral nerve cords and into ectopic brain branches. See also *Figure 2—figure supplement 1A*. Ventral up. Scale bar, 100 μm. (**B**) Fluorescent in situ hybridization (FISH) showing *gluR*⁺ (dd_16476⁺) neurons in ectopic brain branches after 3 weeks of *map3k1* RNAi. Scale bars, 200 μm; higher magnification scale bar, 100 μm. (**A, B**) Ventral up. (**C**) FISH images showing no change in *cintillo*⁺ neuron distribution in the head, but expansion of dd_17258⁺ neurons along the entire AP axis after 3 weeks of *map3k1* RNAi. See also *Figure 2—figure supplement 1B*, (**C**) Ventral up. Scale bars, 200 μm. (**D**) Graph showing no significant difference in the number of dd_17258⁺

*Figure 2 continued on next page*

*Figure 2 continued*

neurons in the heads (AP_1) of *map3k1* and control RNAi animals (p = 0.410; Mann–Whitney test) but a significant difference in the number of ectopic cells observed along the entire AP axis (AP_2 → AP_6: ***p < 0.0006; multiple Mann–Whitney tests). Counted animals underwent 3–4 weeks of RNAi; two replicates were used. No ectopic *cintillo*[+] neurons were observed for both control and *map3k1* RNAi animals. (E) FISH images of unaffected (dd_9223) and affected (dd_7131 and dd_8476) parenchymal cell types in *map3k1* RNAi animals (3 weeks of RNAi). Ventral up. Scale bar, 200 µm. Right graph shows more dd_7131[+] (p < 0.0001; negative binomial regression) and dd_8476[+] (p < 0.0001; negative binomial regression) cells in the tails of *map3k1* RNAi animals compared to control RNAi animals. Counted animals underwent 3–4 weeks of RNAi; three replicates each. Bottom left numbers indicate the number of animals with the result displayed in the image out of the total number of animals observed.

The online version of this article includes the following figure supplement(s) for figure 2:

**Figure supplement 1.** *map3k1* RNAi results in posterior differentiation of some neural cell types.

specification itself occurred in ectopic locations following *map3k1* RNAi. dd_554 transcripts mark a post-mitotic pharynx progenitor population (*Zhu et al., 2015*). Ectopic dd_554[+] cells were present anterior to the pharynx in *map3k1* RNAi animals, including in foci (*Figure 3D*; *Figure 3—figure supplement 1D*). *FoxA*[+]; *smedwi-1*[+] cells prominently, but not necessarily exclusively, include pharynx progenitors. These cells are predominantly centrally restricted on the AP axis to the trunk (*Adler et al., 2014*; *Scimone et al., 2014a*). We observed a small frequency of *FoxA*[+]; *smedwi-1*[+] cells near the brain in both control and *map3k1* RNAi animals (*Figure 3E*, *Figure 3—figure supplement 1E, F*). These findings are consistent with the possibility that these *FoxA*[+]; *smedwi-1*[+] cells include pharynx progenitors, and that ectopic differentiation of pharynx progenitors can occur outside of the predominant pharynx progenitor specification zone at a low frequency.

Complete posterior duplication of the pharynx has previously been observed following *ptk7*, *ndl-3*, and *wntP-2* RNAi (*Scimone et al., 2016*; *Hill and Petersen, 2018*), anterior duplication has been observed after *roboA* RNAi (*Cebrià et al., 2007*), and lateral duplication has been observed after *wnt5* RNAi (*Gurley et al., 2010*). The *map3k1* RNAi phenotype differs from these other patterning phenotypes, in that it involves greater disorganization, with the appearance of small clusters of pharyngeal cells and even single pharyngeal cells instead of only well-organized but ectopically placed pharynges. This scenario bears similarity to the phenotype for the eye: disorganized ectopic tissue differentiation, including in small clusters and single cells. These findings are consistent with the possibility that *map3k1* RNAi disrupts the regulation of progenitor targeting that normally results in differentiation being restricted to occur in precise locations.

## The targeting and maintenance of tissues after *map3k1* RNAi

Post-mitotic progenitors normally migrate to, and differentiate at, their TZ. Prior work indicates that in addition to the TZ, the target tissue itself can incorporate its fate-specified progenitors and promote progenitor differentiation, even at an ectopic location (*Atabay et al., 2018*; *Hill and Petersen, 2018*). For instance, surgically transplanting an ectopic eye outside of the TZ, but within the broad distribution of eye progenitors, results in a stable ectopic eye that incorporates progenitors to replace dying cells as part of turnover (*Atabay et al., 2018*). This is enabled by the fact that progenitors in wild-type animals are specified in broad regions, giving ectopic differentiated tissues access to a constant supply of progenitors (*Lapan and Reddien, 2011*; *Lapan and Reddien, 2012*). Thus, at least two system components appear to be capable of promoting progenitor differentiation: the TZ and the target tissue. The pharynx shows similar properties to the eye – with an ectopic pharynx being maintained through progenitor incorporation and differentiation (*Hill and Petersen, 2018*). An ectopic organ, however, will not regenerate upon its removal. Because the original TZ location of the tissue is unchanged in this situation, progenitors will target the correct location after resection of an ectopic organ (*Atabay et al., 2018*; *Hill and Petersen, 2018*).

Given the above reasoning, ectopic eyes in *map3k1* RNAi animals could, in principle, be explained by TZ movement or expansion. To assess whether TZ alteration occurred following *map3k1* RNAi, or some other explanation for the *map3k1* RNAi phenotype is more likely, we resected all visible eyes in *map3k1* RNAi animals to determine the location of new progenitor differentiation in the absence of a target tissue. Eye-resected *map3k1* RNAi animals regenerated eyes at a comparable rate to control animals, and in the normal TZ location (*Figure 4A*; *Figure 4—figure supplement 1A*). Notably, however, *map3k1* RNAi animals did not regenerate any resected ectopic eyes (*Figure 4A*). This indicates that the TZ location is maintained after *map3k1* RNAi, and that at least some progenitors are

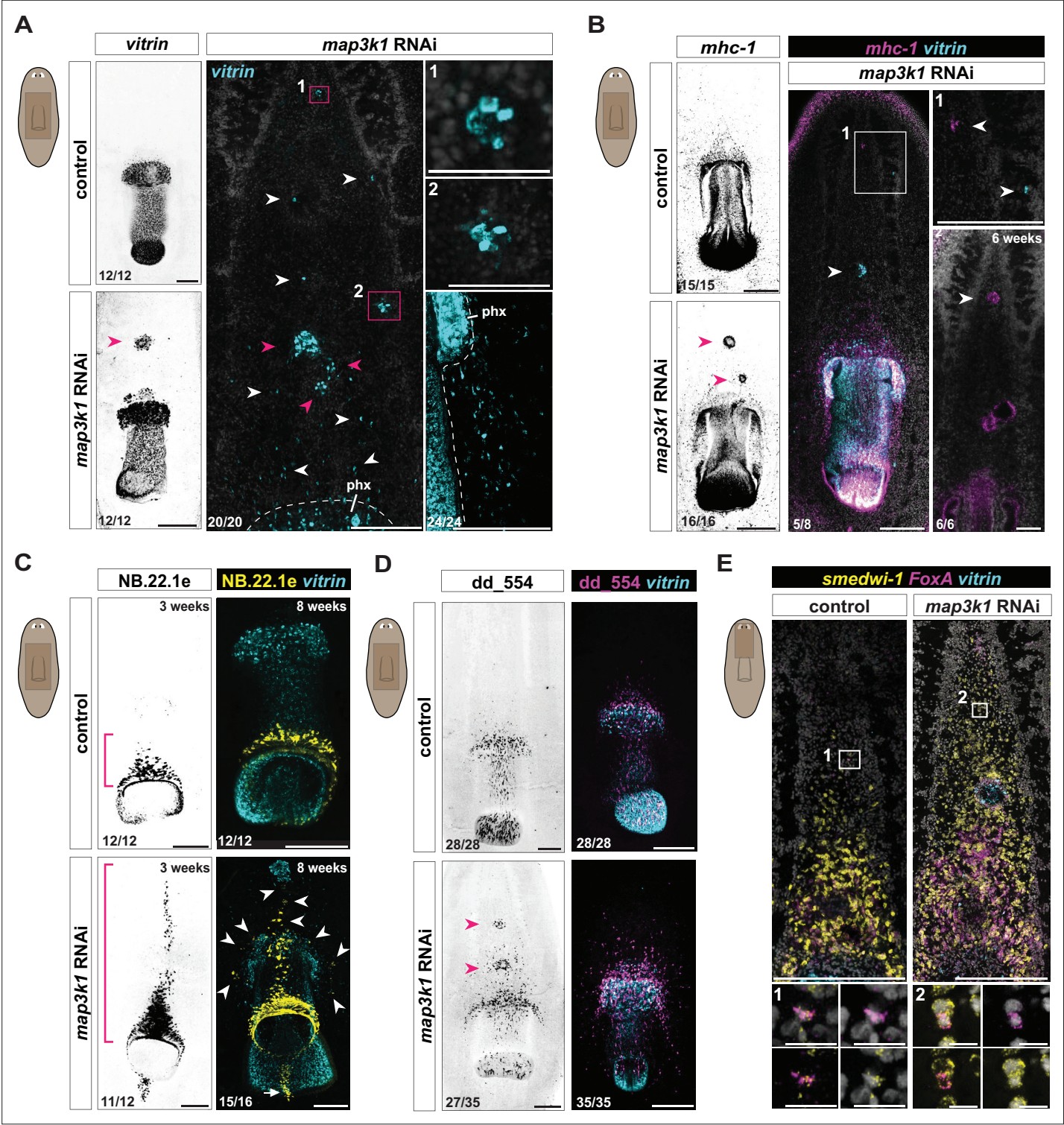

**Figure 3.** *map3k1* RNAi results in pharynx cell types in ectopic anterior locations. (**A**) Fluorescent in situ hybridization (FISH) images of control and *map3k1* RNAi animals showing anterior expansion and dispersal of *vitrin*⁺ (pharynx) single cells (white arrows) and clusters of cells (pink arrows and boxes) at variable positions along the AP and ML axes at 3 weeks (*n* = 20/20; two replicates) and 4 weeks (*n* = 12/12; one replicate) of RNAi, between the cephalic ganglia (1), near the ventral nerve cords (2), and lateral to the pharynx (lower right panel). Scale bars, 200 µm; magnified images scale bars, 20 µm. Control animals, 3 weeks of RNAi. (**B**) Left panels: FISH images showing anterior clusters of *mhc-1*⁺ (pharynx muscle) cells (pink arrows) in *map3k1* RNAi animals. The middle panel shows ectopic *mhc-1*⁺ cells and *vitrin*⁺ cells near the brain in *map3k1* RNAi animals (*n*=5/8; 1 replicate, white arrows). Left and middle panels, 3 weeks RNAi. Bottom right panel shows ectopic large clusters of *mhc-1*⁺ cells between the cephalic ganglia after 6 weeks of

*Figure 3 continued on next page*

*Figure 3 continued*

RNAi (*n* = 6/6; 1 replicate, white arrows). Ventral up. Scale bar, 100 μm. (**C**) FISH images of *map3k1* RNAi animals showing NB.22.1e⁺ mouth cells anterior to the normal location (pink brackets; 3 weeks RNAi) and dispersed around the pharynx (white arrows; 8 weeks RNAi). Scale bar, 100 μm. (**D**) FISH images showing clusters of dd_554⁺ cells (white arrows) (intermediate pharynx progenitor population; *Zhu et al., 2015*) and single dd_554⁺ cells dispersed around and anterior to the pharynx (*vitrin*) in 3-week *map3k1* RNAi animals. Counted animals for all panels underwent 3–4 weeks of RNAi; three replicates were used. Ventral up. Scale bar, 200 μm. (**E**) FISH showing examples of *FoxA⁺; smedwi-1⁺* cells in the head region of both control and *map3k1* RNAi animals (white boxes). Three weeks RNAi, two replicates. See also *Figure 3—figure supplement 1E*. Scale bars, 200 μm; magnified image scale bars, 10 μm (**A–D**). All panels, ventral up. Numbers in each panel indicate the number of animals displaying the result shown in the image out of total animals observed.

The online version of this article includes the following figure supplement(s) for figure 3:

**Figure supplement 1.** *map3k1* RNAi causes ectopic differentiation of pharyngeal cell types.

capable of reaching the normal TZ location in the absence of an eye. These data are consistent with a model in which *map3k1* does not primarily control the positional information read by progenitors, but instead affects the ability of progenitors to differentiate at proper locations in response to a normal positional information system – a possibility explored further below.

To determine if the pharynx TZ is also maintained in its normal, wild-type location in *map3k1* RNAi animals, we removed the entire pharynx of *map3k1* RNAi and control animals (*Figure 4B*, *Figure 4— figure supplement 1B*). By 10 days after pharynx resection, animals regenerated a pharynx in the original position, but with some disorganization and multiple pharyngeal structures forming in some cases (*Figure 4B*, *Figure 4—figure supplement 1B, C*). Thus, the case for the pharynx is more complex than for the eye. Regardless, these findings for the pharynx suggest that at least the normal central TZ remains and is not simply shifted anteriorly.

## The location of de novo organ regeneration after *map3k1* RNAi

To determine the location of progenitor targeting and de novo organ formation during regeneration, we amputated *map3k1* RNAi animals into head, trunk, and tail fragments and analyzed them after 10 days of regeneration (*Figure 4C*). All fragments regenerated organs that they did not already contain at the time of amputation (*Figure 4D*). Two eyes formed in approximately normal locations in *map3k1* RNAi head blastemas, rather than being posteriorly shifted or appearing in multiple locations initially. This suggests, like the results from the eye-resection experiments described above, that the TZ is regenerated in roughly the wild-type location in *map3k1* RNAi animals.

Prior work in the planarian species *D. japonica* showed that *map3k1* RNAi results in tail fragments regenerating pharynges in an anterior-shifted location (*Hosoda et al., 2018*). Consistent with this previously reported effect, pharynges regenerated more anteriorly in *map3k1* RNAi *S. mediterranea* tail fragments, just posterior to the regenerating brain (*Figure 4C–E*). Notably, trunks also regenerated secondary pharynx-like aggregates very close to the brain that were underdeveloped and that appeared to interfere with structures around them (*Figure 4D*). Head fragments regenerated multiple pharyngeal structures (*Figure 4—figure supplement 1D*). Therefore, the situation for the pharynx in regeneration is more complex than that of the eye, similar to the findings for organ resections described above.

## PCG expression domains are largely unaffected by *map3k1* RNAi

As noted above, numerous PCGs can be inhibited to cause organ duplications. However, the *map3k1* RNAi phenotype described so far is largely consistent with alteration of progenitor targeting behavior rather than global shifting of positional information. To directly examine the spatial maintenance of positional information in *map3k1* RNAi animals, we labeled these animals with RNA probes for multiple PCGs. Anterior, posterior, and medial PCG expression domains were largely unaffected in *map3k1* RNAi animals – including expression domains for *sFRP-1*, *ndl-4*, *ndl-5*, *ndl-2*, *ndl-3*, *wntP-2*, *axinB*, *sp5*, *ptk7*, *wnt11-1*, and *slit* (*Figure 5A*). The spatial distribution of *axinB* transcription – a read-out of the posterior-to-anterior Wnt activity gradient in planarians – was similar in control and *map3k1* RNAi animals, indicating that Wnt activity was maintained regionally on the AP axis (*Figure 5A*; *Iglesias et al., 2011*; *Reuter et al., 2015*; *Stückemann et al., 2017*; *Tewari et al., 2019*).

The *notum⁺* anterior pole, which is produced from neoblast-derived and migratory *FoxD⁺* progenitors (*Roberts-Galbraith and Newmark, 2013*; *Scimone et al., 2014b*; *Vogg et al., 2014*), was present,

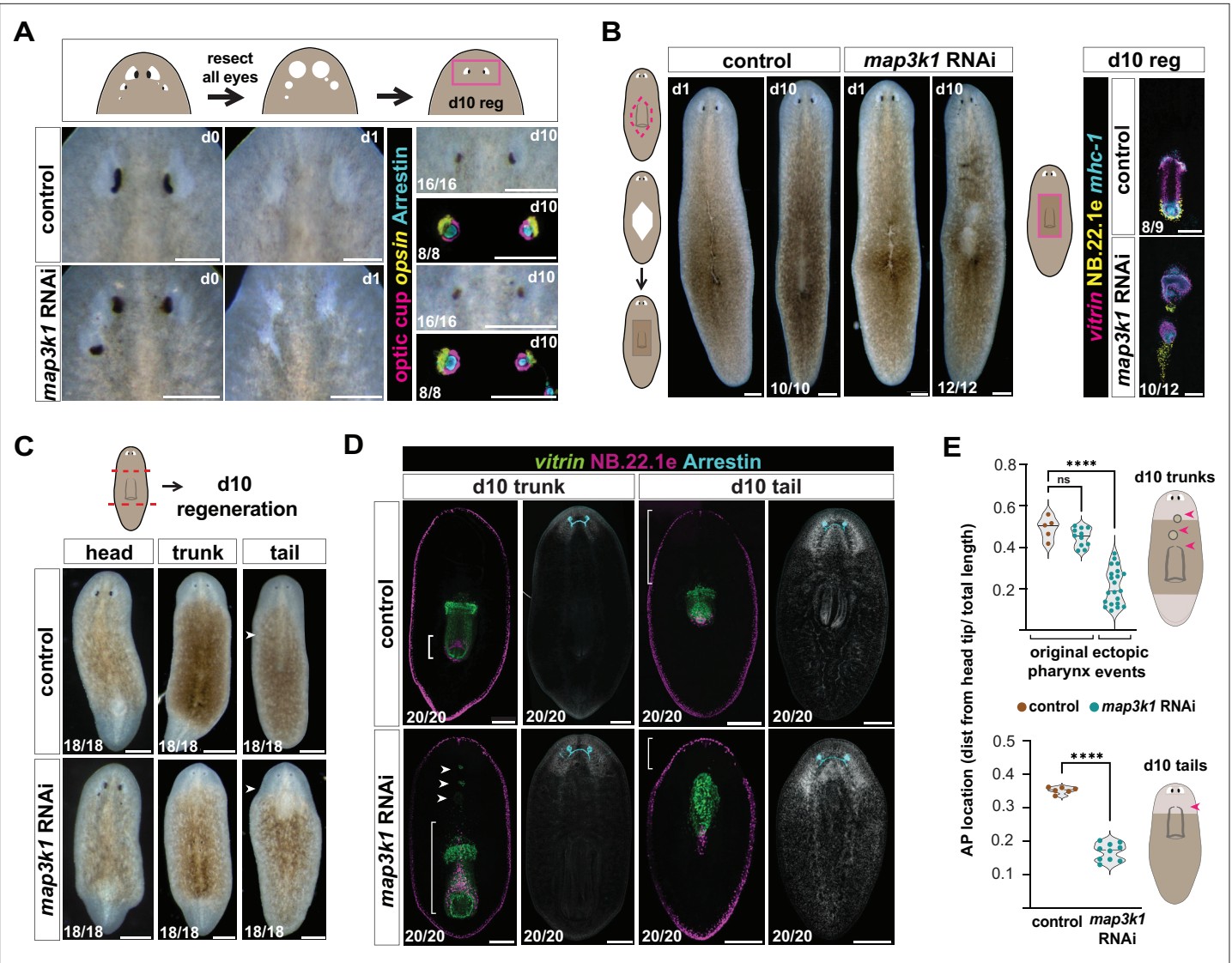

**Figure 4.** *map3k1* RNAi animals display tissue-specific regeneration at target zones. (**A**) Top schematic depicts eye resection experimental design. Live images of control and *map3k1* RNAi animals prior to (d0; left column) and the day after (d1; middle column) eye resection. The right column shows live and fluorescent in situ hybridization (FISH) images of control and *map3k1* RNAi animals 10 days following eye resection. Photoreceptor neurons (PRNs) are visualized with anti-Arrestin (PRN cell bodies and projections) and an RNA probe to *opsin* (PRN cell bodies). Optic cup cells are visualized using a pool of *catalase1/tyrosinase/glut3* RNA probes. Three weeks of RNAi was performed prior to resections; FISH and live images are from different animals. Dorsal, up. Scale bars, 200 µm. See *Figure 4—figure supplement 1A* for d0 FISH. (**B**) Diagram of pharynx resection on the left. Live images at d1 and d10 after pharynx resection showing pharynx regeneration in the correct location for both *map3k1* RNAi and control animals. Pharynx regeneration in *map3k1* RNAi animals is disorganized; FISH contains RNA probes to *vitrin* (pharynx-specific), NB.22.1e (mouth and esophagus), and *mhc-1* (pharynx muscle). See also *Figure 4—figure supplement 1B* for d0 FISH. Ventral up. Live images, scale bars, 200 µm; FISH images, scale bars, 100 µm. (**C**) Live image of d10 head, trunk, and regenerating tail fragments. *map3k1* RNAi tail fragments regenerate pharynges (white arrow) in a more anterior location compared to control animals. (**C, D**) Three weeks of RNAi was performed prior to resection. Scale bar, 100 µm. (**D**) FISH images showing anterior *vitrin*⁺ pharynx cells (white arrows) and NB.22.1e⁺ mouth cells (white brackets) in *map3k1* RNAi day 10 trunk regenerates. All *map3k1* RNAi trunks and tails fully regenerate eyes (anti-Arrestin). Day 10 tails regenerate pharynges in a more anterior location compared to control animals. Ventral up. Scale bar, 200 µm. (**E**) Top graph shows no difference in the AP location of original pharynges in control and *map3k1* RNAi d10 trunk regenerates (p = 0.1054; Welch's *t*-test) but a significant difference between control original pharynges and ectopic pharyngeal cell clusters *map3k1* RNAi trunks (****p < 0.0001; Mann–Whitney test). The bottom graph shows an anterior shift in pharynx regeneration in *map3k1* RNAi tail fragments compared to control animals (p < 0.0001; Welch's *t*-test). Numbers in each panel indicate number of animals displaying the result shown in the image out of total animals observed.

The online version of this article includes the following figure supplement(s) for figure 4:

**Figure supplement 1.** *map3k1* RNAi animals undergo tissue-specific and whole-body regeneration, with some errors in pharynx regeneration.

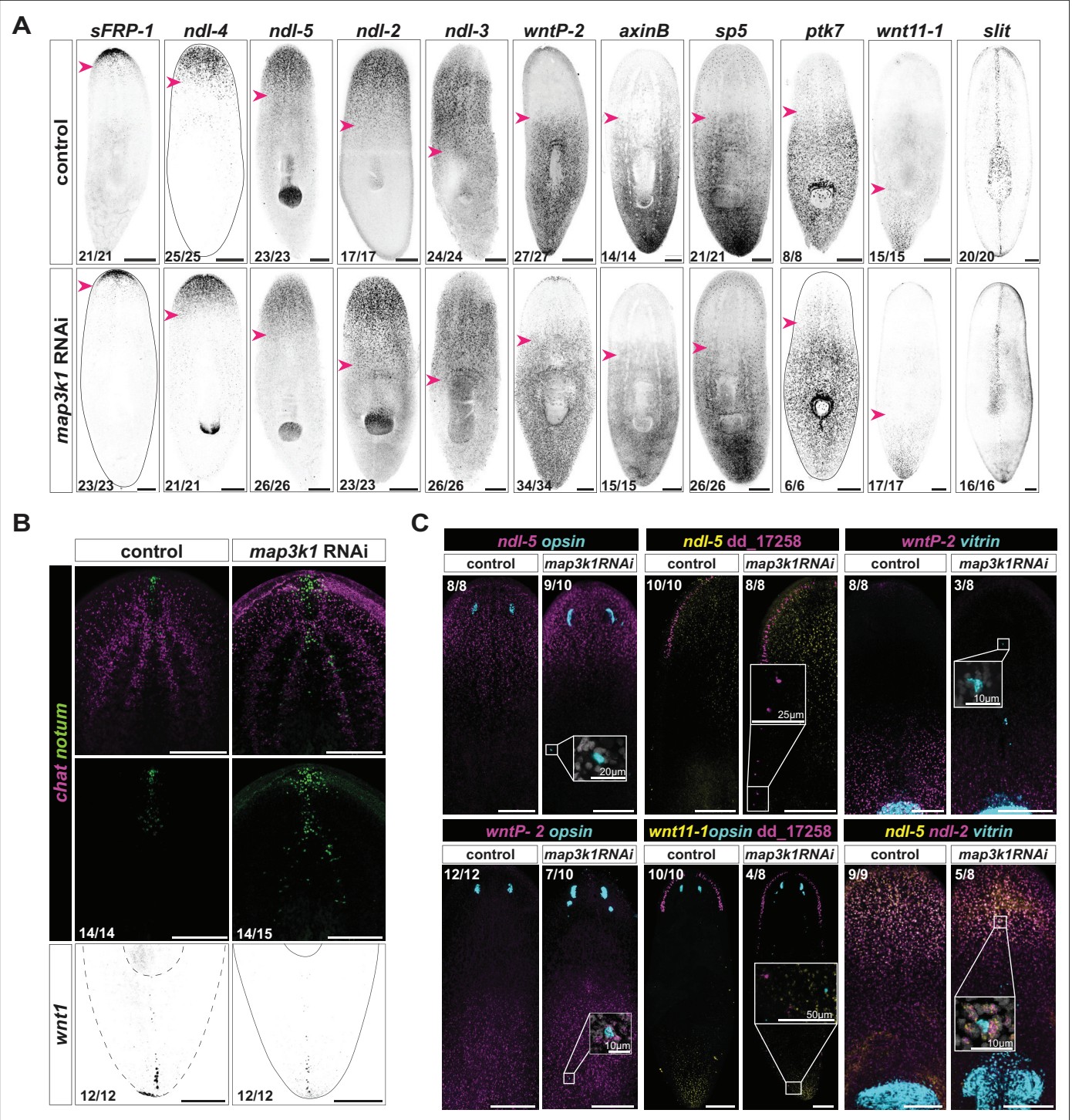

**Figure 5.** Positional information remains largely unaffected in *map3k1* RNAi animals. (**A**) Fluorescent in situ hybridization (FISH) panel of position control gene (PCG) expression (*sFRP-1*, *ndl-4*, *ndl-5*, *ndl-2*, *ndl-3*, *wntP-2*, *axinB*, *sp5*, *ptk7*, *wnt11-1*, and *slit*) shows no obvious changes to expression domains in *map3k1* RNAi animals. Animals from 3 and 4 weeks of RNAi were used. Pink arrowheads mark the end of PCG expression domains. Scale bars, 200 μm. Ventral up. (**B**) FISH example showing some dispersion of *notum*+; *chat*+ cells in the brain of a *map3k1* RNAi animal after 3 weeks of RNAi (sample numbers from 3 and 4 weeks of RNAi). No obvious changes in *wnt1*+ posterior pole organization were observed after 2 and 3 weeks of RNAi (example FISH, 2 weeks RNAi). Scale bars, 200 μm. Ventral up. (**C**) Top row panels show example FISH images of *opsin*+ (photoreceptor neurons, PRNs), dd_17258+ (neuron type), and *vitrin*+ (pharynx) cells outside of typical PCG expression domains (*ndl-5*, *ndl-5*, and *wntP-2*, respectively). Bottom row panels show example FISH images of *opsin*+, dd_17258+, and *vitrin*+ cells inside PCG expression domains they normally are not found in (*wntP-2*, *wnt11-1*, and *ndl-5/*

*Figure 5 continued on next page*

*Figure 5 continued*

*ndl-2*, respectively). Far right panels on top and bottom, ventral up. All other panels, dorsal up. Animals from 3 and 4 weeks of RNAi were used. Scale bars, 200 μm. Numbers in each panel indicate number of animals displaying the result shown in the image out of total animals observed.

The online version of this article includes the following figure supplement(s) for figure 5:

**Figure supplement 1.** *map3k1* RNAi tail fragments can regenerate the anterior pole but regenerate their new pharynx at a more anterior location.

**Figure supplement 2.** *map3k1* is expressed in neoblasts and post-mitotic progenitors.

but displayed some dispersal of cells after *map3k1* RNAi (*Figure 5B*; *Figure 5—figure supplement 1A*). *notum⁺*; *chat⁺* brain cells in the cephalic ganglia also appeared more dispersed in *map3k1* RNAi animals, and this pattern was somewhat reminiscent of the eye and pharynx phenotypes. *notum⁺* cell disorganization was also exaggerated during tail regeneration, when the animal is challenged with generating a new head (*Figure 5—figure supplement 1B*). Despite some anterior pole dispersal, the posterior pole appeared mostly normal and regenerating tails were still able to obtain proper PCG expression domains by day 4 (*Figure 5B*; *Figure 5—figure supplement 1C*).

Ectopic tissues in *map3k1* RNAi animals formed outside of the PCG expression domains that they are normally restricted to; for instance, *opsin⁺* and dd_17258⁺ neurons formed outside of the *ndl-5⁺* domain and *vitrin⁺* pharyngeal cells formed outside of the *wntP-2⁺* domain (*Figure 5C*). Ectopic differentiated cells were also observed inside PCG expression domains that they are usually not found within; for instance, *opsin⁺* and dd_17258⁺ neurons formed within a domain expressing *wntP-2* and *wnt11-1*, and *vitrin⁺* cells formed within *ndl-2 and ndl-5* expression domains (*Figure 5C*). In *map3k1* RNAi tail fragments, pharynges regenerated partially outside of the normal *wntP-2⁺* expression domain, despite *wntP-2* having a known role in establishing trunk identity (*Figure 5—figure supplement 1D*). These findings are consistent with the results described above: eyes are maintained and regenerated at the normal TZ, indicating that the defining positional information for the TZ remains at approximately the correct location in *map3k1* RNAi animals. The fact that ectopic eyes emerge over time in a disordered fashion suggests that it is progenitor targeting for differentiation at precise locations that is affected by *map3k1* RNAi rather than positional information itself.

## *map3k1* is expressed in neoblasts and migratory post-mitotic progenitors

Recent scRNA-seq work has annotated fate-associated clusters of planarian neoblasts and post-mitotic progenitors (*King et al., 2024*). If *map3k1* acts in progenitors to regulate their differentiation, it should be transcribed in these cells. Indeed, *map3k1* transcripts were present broadly across neoblast and post-mitotic progenitor clusters (*Figure 5—figure supplement 2A, B*), including for the eye, neural classes affected by *map3k1* RNAi, and the pharynx (*Figure 5—figure supplement 2C–I*). Whereas expression data alone do not necessarily indicate the site of action of *map3k1*, the data are consistent with the possibility that *map3k1* can act within migratory progenitors.

## *map3k1* is required for restricting differentiation of eye progenitors along their migratory path to the TZ

Eye and pharynx progenitors are normally specified in restricted domains (*Lapan and Reddien, 2011*; *Lapan and Reddien, 2012*; *Adler et al., 2014*; *Scimone et al., 2014a*; *Atabay et al., 2018*), referred to here as progenitor specification zones. Within these domains, progenitors can migrate to reach their target tissue at their TZ, where they differentiate (*Atabay et al., 2018*; *Hill and Petersen, 2018*). We sought to test the possibility that ectopic cell differentiation, in *map3k1* RNAi animals, is a result of premature progenitor differentiation at positions along the normal migratory path before reaching the TZ. An alternative scenario we considered is that ectopic progenitor specification at some distant location occurs after *map3k1* RNAi and requires excessively long-range progenitor migration to reach the TZ, ultimately resulting in ectopic differentiation.

Neoblasts can be killed with irradiation (*Bardeen and Baetjer, 1904*), and lead shielding can be utilized in X-irradiation experiments to locally protect neoblasts, resulting in neoblasts being present only in a restricted field (*Dubois, 1948*; *Guedelhoefer and Sánchez Alvarado, 2012*; *Abnave et al., 2017*). Neoblasts expand slowly from these restricted regions (barring amputation) (*Saló and Baguñà, 1985*), but their postmitotic descendant cells serve as precursors that can readily migrate to target

tissues (*Guedelhoefer and Sánchez Alvarado, 2012*; *Abnave et al., 2017*; *Park et al., 2023*). We utilized lead shielding over the top half of the animal to preserve the region of eye neoblasts, the place where eye progenitors are normally born, from X-irradiation (*Figure 6A*). Forty-eight hours after irradiation, we initiated RNAi of *map3k1* to observe the behavior of eye progenitors – restricted to be born within their normal, primary eye-progenitor specification zone (the zone of surviving neoblasts). Twelve to fourteen days after the initiation of *map3k1* RNAi in these partially irradiated animals, ectopic eye cells and ectopic dd_17258[+]neurons were apparent (*Figure 6A*, *Figure 6—figure supplement 1A, B*). This indicates that ectopic differentiation can occur after *map3k1* RNAi even from neoblasts restricted to undergo fate specification in the normal location.

We next sought to confirm that the appearance of ectopic cells required the local production of new progenitors from neoblasts. We utilized lead shielding over the posterior half of wild-type animals and performed X-irradiation to preserve neoblasts only in the tail region (*Figure 6B*). We then performed *map3k1* RNAi over a 10- to 12-day period and assessed animals for ectopic cell differentiation. The AP location of ectopic cells in these animals was most frequently in the area of remaining neoblasts (labeled with a probe to *smedwi-1* transcript), with a small frequency of ectopic differentiation events occurring outside of this area, but still proximal to it (*Figure 6B*).

To assess the behavior of *map3k1* RNAi eye progenitors in the normal eye-progenitor specification zone and in a non-RNAi host environment, we labeled neoblasts with EdU after 2–3 weeks of *map3k1* RNAi, then transplanted a small pre-pharyngeal tissue fragment from these animals into the pre-pharyngeal region of unlabeled, wild-type host animals (hosts had not experienced *map3k1* RNAi) (*Figure 6C*). Transplant recipient animals were then fixed 12 days later. EdU-positive cells migrated out of the transplantation region and differentiated into eye cells at both the normal eye location (at the TZ) and at ectopic locations in the host environment (*Figure 6C, D*; *Figure 6—figure supplement 1C*). Additional ectopic eye cells were observed following transplantation that were not EdU-positive, likely because of incomplete labeling or because they were born multiple divisions after the EdU pulse was delivered (*Figure 6—figure supplement 1D*). These observations indicate that eye progenitors originating from the normal specification zone can erroneously differentiate before reaching their target location, including in a non-RNAi host environment.

## Differentiated cells in the wrong organ of map3k1 *RNAi* animals

We administered *map3k1* dsRNA feedings for RNAi and let animals undergo normal long-term tissue turnover to observe the consequences of errors in progenitor targeting on tissue pattern. Eight weeks of *map3k1* RNAi led to differentiated pharynx muscle (*mhc-1*) cells within the cephalic ganglia (*Figure 7A*; *Figure 7—figure supplement 1A*). Single epidermal (NB.22.1e) cells were also observed, at a low frequency ($n = 4/18$), within eyes after 8 weeks of *map3k1* RNAi. Tissue incorporation errors did not require many RNAi feedings to manifest. After only 3 weeks of *map3k1* RNAi and tail amputation, day 10 tail regenerates ($n = 9/24$) displayed ectopic *vitrin*[+] pharyngeal cells in eyes (*Figure 7B*; *Figure 7—figure supplement 1B*). These striking cell-organization defects were not previously observed in the patterning phenotypes of PCG RNAi animals. We suggest that this defect highlights the risk to tissue architecture of not enacting tight spatial regulation of differentiation, especially in a biological context where progenitors are spatially dispersed and migratory. Ectopic differentiated cells can become incorporated into inappropriate tissue environments where they would normally never be observed, potentially disrupting tissue structure and function.

## *map3k1* RNAi animals develop teratomas

All *map3k1* RNAi animals ultimately developed tissue growths within 8–12 weeks, simply from undergoing tissue turnover without injury (*Figure 7C*; *Figure 7—figure supplement 1C*). These growths predominantly formed in the anterior of the animal and presented as a heterogeneous collection of cell and tissue types and consistently contained clusters of neurons and muscle cells (*Figure 7D*; *Figure 7—figure supplement 1D*). Pharynx (*vitrin*[+]) cells and gland cells (*mag-1*[+]) were not as common in lateral outgrowths (*Figure 7D*). Cell types that were present in these growths included eye cells, neurons from the central nervous system (*cintillo*[+], dd_17258[+], and dd_3524[+] cells), glia, muscle cells, and epidermal cells (*Figure 7D*). We considered these aberrant growths with a heterogeneous collection of intermingled tissues to be teratomas. A similar teratoma formation defect has been observed in planarians with a defect in progenitor migration caused by *integrin* RNAi (*Bonar and Petersen,*

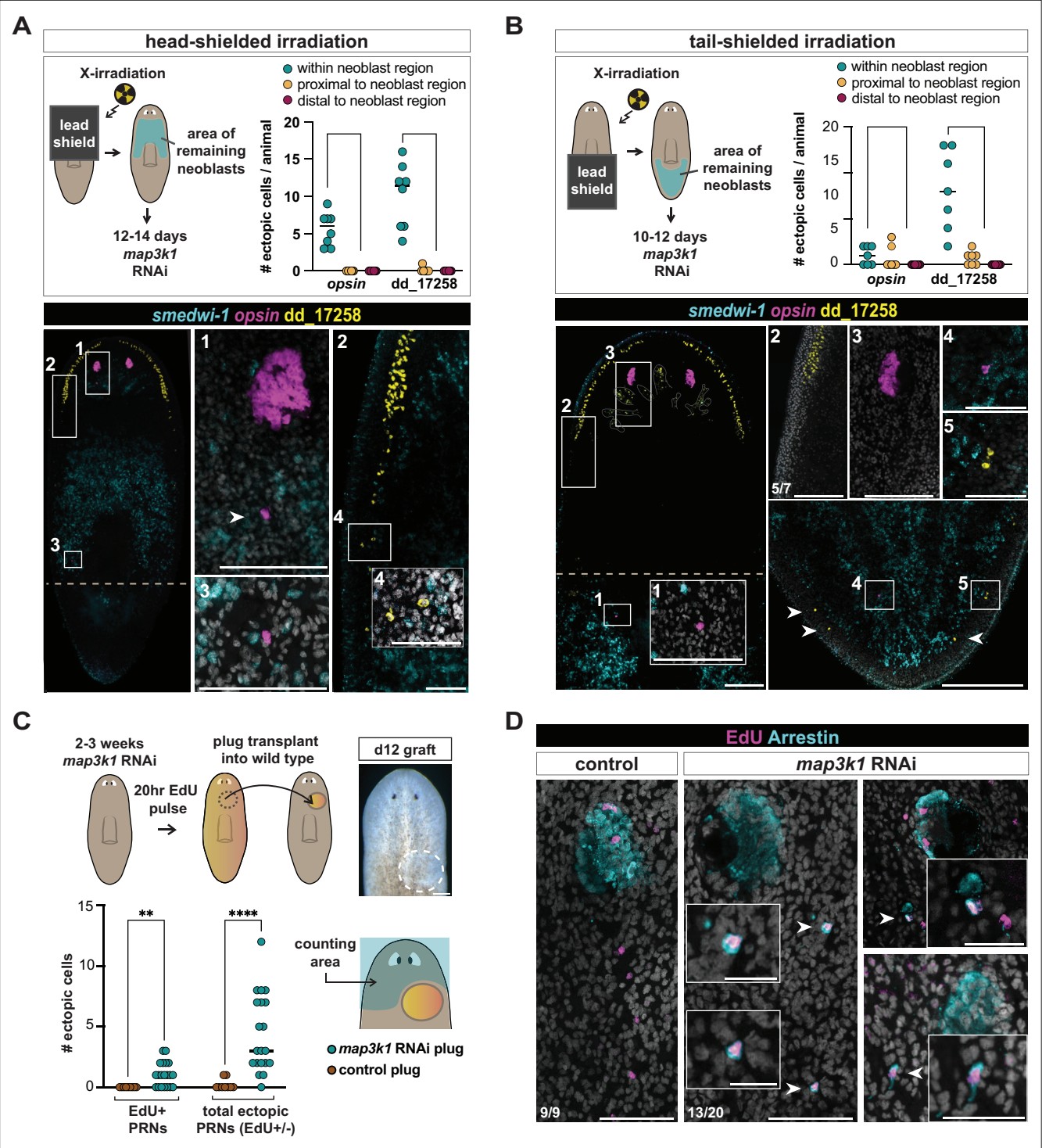

**Figure 6.** *map3k1* RNAi eye progenitors prematurely differentiate along a normal migratory path. (**A**) Schematic of head-shielded irradiation experimental design. Animals were fixed 12–14 days following the first *map3k1* RNAi feeding (2 days after irradiation), depending on health by inspection. The right graph shows ectopic differentiation events were more likely to occur in the anterior half of the animal (surviving neoblast region; marked by a *smedwi-1+* RNA probe) versus outside of (distal or proximal to) the neoblast region for both *opsin+* and dd_17258+ neurons (p < 0.0001; binomial exact test). All observed ectopic photoreceptor neurons (PRNs, *opsin*) and dd_17258+ neurons outside of the neoblast region were present proximal to (within 50 μm) the neoblast region. Bottom fluorescent in situ hybridization (FISH) panels depict examples of ectopic PRN (anti-Arrestin) and dd_17258+ neuron differentiation events within the neoblast region. Scale bar, 200 μm; magnified boxes 1–4, scale bars, 50 μm. (**B**) Schematic of tail-shielded irradiation experimental design. Animals were fixed 10–12 days following the first *map3k1* RNAi feeding (2 days after irradiation), depending

*Figure 6 continued on next page*

*Figure 6 continued*

on health by inspection. The right graph shows ectopic events were more likely to occur in the tail (area of surviving neoblasts) versus outside of (distal or proximal to) the neoblast region for dd_17258⁺ neurons (p < 0.0001; binomial exact test). All observed ectopic PRNs dd_17258⁺ neurons outside of the neoblast region were present proximal to (within 50 µm) the neoblast region. FISH panels depict examples (white arrows) of ectopic PRNs (*opsin*) and dd_17258⁺neurons in the tail of a tail-shielded, irradiated *map3k1* RNAi animal. Scale bars, 200 µm; magnified boxes 1–5, scale bars, 50 µm. (**C**) Schematic of EdU-labeled graft transplant experimental design; bottom graph shows a significant number of total ectopic eye cells (****p < 0.0001; Mann–Whitney test) and ectopic EdU-positive eye cells (**p = 0.002; Mann–Whitney test) in recipient wild-type animals compared to control. Animals after 2 and 3 weeks of RNAi prior to the EdU pulse were used for transplantation. (**D**) FISH example of EdU-positive ectopic eye cells (white arrows) differentiated in wild-type animals (*n* = 13/20) with EdU-positive ectopic cells; *n* = 19/20 exhibited any ectopic eye cells outside of the transplant area. Scale bars, 200 µm; zoom in scale bars, 20 µm. (**A–D**) All panels are dorsal up. Numbers in each panel indicate the number of animals displaying the result shown in the image out of total animals observed.

The online version of this article includes the following figure supplement(s) for figure 6:

**Figure supplement 1.** *map3k1* RNAi animals display ectopic differentiation of local neoblasts.

*2017*; *Seebeck et al., 2017*). It is known that differentiated tissues, such as the eye, can trap their own progenitors and lead to their differentiation (*Atabay et al., 2018*; *Hill and Petersen, 2018*). This suggests that ectopic differentiation in inappropriate locations in *map3k1* RNAi animals can result in the trapping of additional progenitors and can ultimately lead to an inappropriate aggregate of differentiated cells with aberrant pattern and organization. This defect highlights a further risk to tissue architecture if targeting and differentiation of migratory progenitors is not tightly controlled.

## Discussion

Planarians display continuous turnover of adult tissues through the fate specification and differentiation of adult stem cells called neoblasts (*Reddien, 2018*). Fate specification in neoblasts can occur regionally (such as in the head for eye neoblasts) but is still spatially broad and intermingled (*Park et al., 2023*). Fate-specified neoblasts (specialized neoblasts) produce progeny cells that serve as precursors, referred to here as post-mitotic progenitors. These post-mitotic progenitors migrate to target locations to produce highly patterned anatomy. We suggest that differentiation is restricted during migratory targeting as an essential component of pattern formation, with the *map3k1* RNAi phenotype indicating the existence and purpose of this element of patterning (*Figure 8*). We further suggest that progenitor targeting for local differentiation requires regulation from stem-cell-extrinsic signals, in the form of regionally expressed genes in muscle that comprise adult planarian positional information. How positional information interfaces with neoblasts and post-mitotic progenitors at the molecular level to regulate migratory assortment and differentiation only at precise locations is a fundamental problem of planarian regeneration and progenitor differentiation regulation. We suggest a model in which *map3k1* acts as a brake on differentiation in stem cell progeny that can be lifted when suitable differentiation cues are encountered, either in the form of positional information (a TZ) or from interaction with target tissues (*Figure 8A*). We suggest that *map3k1* is not required for the spatial distribution of progenitor-extrinsic differentiation-promoting cues themselves, but for progenitors to be restricted from differentiating until these cues are received (*Figure 8B*). Independent work in *Lo and Petersen, 2025* showed that *map3k1* perturbation leads to ectopic progenitor differentiation in *S. mediterranea*, providing supporting evidence for the findings presented here.

Several observations support this model. First, PCG expression patterns themselves were largely normal following *map3k1* RNAi. In the planarian species *D. japonica*, *map3k1* RNAi was reported to cause an anterior expansion of *sp5* expression in regenerating tails, but we did not note overt *sp5* expression change in *S. mediterranea*. Additionally, recent work reported a slight posterior expansion of *ndl-5* expression following *map3k1* RNAi (*Lo and Petersen, 2025*), but this shift was small in magnitude. Second, ectopic differentiated cells were present directly within PCG expression domains that they are normally restricted from in *map3k1* RNAi animals. Third, ectopic differentiation after *map3k1* RNAi was more spatially disorganized than is typically observed for patterning phenotypes that occur following the shifting of PCG expression domains. For example, a posterior line of ectopic eyes emerges in *ndk* RNAi animals (*Cebrià et al., 2002*) and a lateral line of ectopic eyes emerges in *wnt5* RNAi animals (*Atabay et al., 2018*), whereas ectopic eye cells emerged across AP and ML axes in a disorganized manner in *map3k1* RNAi animals. Fourth, ectopic cells following *map3k1* RNAi

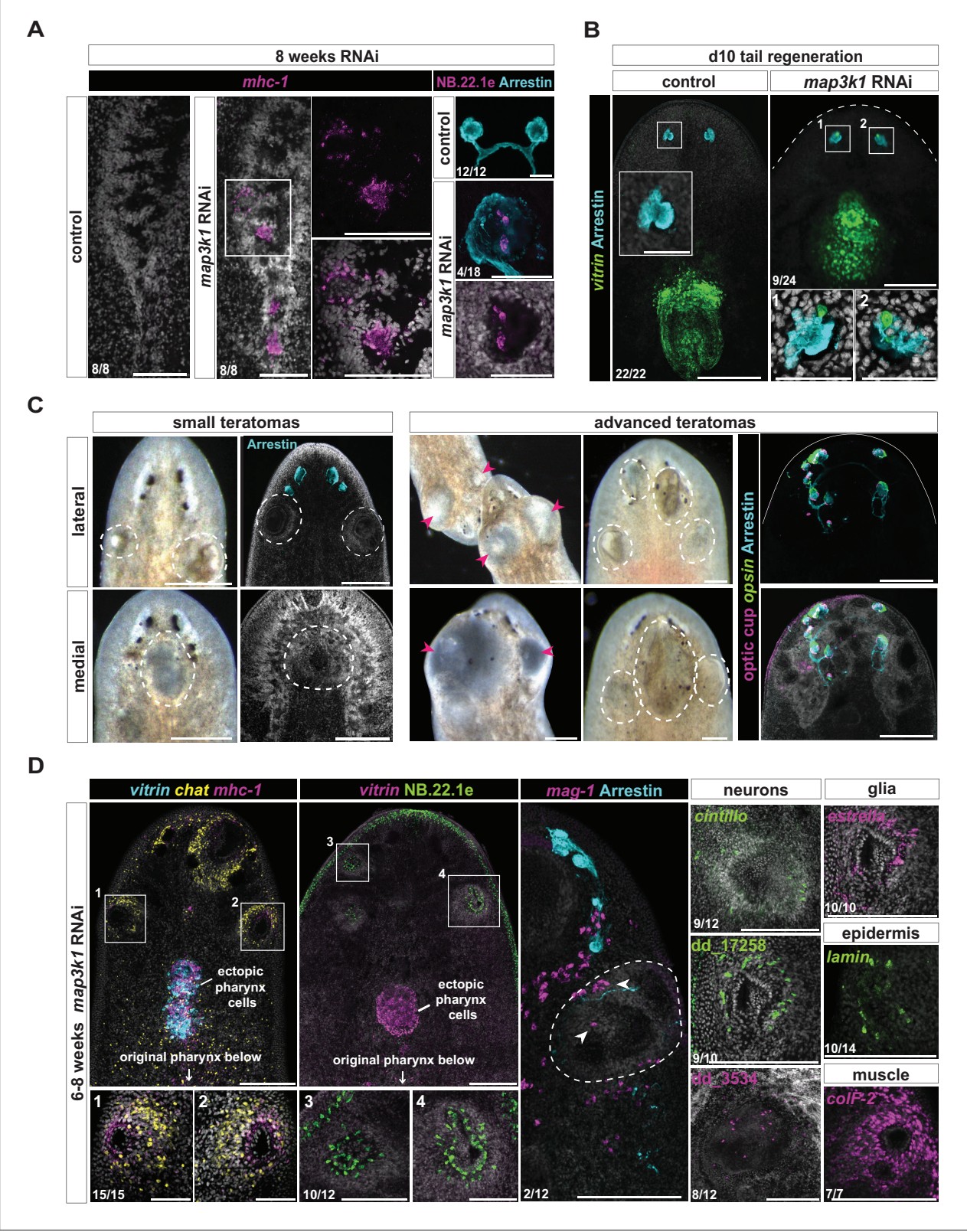

**Figure 7.** *map3k1* RNAi results in progenitor differentiation in incorrect organs and teratoma formation. (**A**) Fluorescent in situ hybridization (FISH) showing clusters of *mhc-1+* cells within the lobes of the brain and ventral nerve cords, and NB.22.1e+ cells (epidermis, mouth) present within the eye after 8 weeks of *map3k1* RNAi. Brain, ventral up; eye, dorsal up. Scale bars, 100 µm. (**B**) FISH examples of *vitrin+* (pharynx) cells present within eyes (anti-Arrestin) at day 10 tail regeneration, following 3 weeks of *map3k1* RNAi. See also *Figure 7—figure supplement 1B*. Dorsal up. Scale bars, 200 µm;

*Figure 7 continued on next page*

*Figure 7 continued*

scale bars for magnified eye images, 50 μm. (**C**) Left panels: live images of small growths in animals at 12 weeks of *map3k1* RNAi feedings, accompanied by DAPI images of similarly positioned growths. Right panels: live images of advanced teratomas (pink arrows) in animals at 12 weeks of *map3k1* RNAi, accompanied by a FISH example showing photoreceptors (*opsin* and anti-Arrestin) and optic cup (OC) cells (*tyrosinase, catalase1, glut3*) scattered in and around the teratomas. Bottom left animal, 8 weeks RNAi. (**C, D**) Scale bars, 200 μm. Dorsal up. (**D**) Left panels: FISH images showing *chat+, mhc-1+,* and NB.22.1e+ cells are common in teratomas, often excluding *vitrin+* (pharynx) and *mag-1+* (gland cells). Scale bars, 100 μm; panels 1–4, scale bars, 50 μm. Right panels show examples of other cell types commonly found in outgrowths: *cintillo+* (neuron), dd_17258+ (neuron), dd_3534+ (neuron), *estrella+* (glia), anti-Arrestin+ (photoreceptors), NB.22.1e+, *lamin+* (mouth and epidermis), and *colF-2+* (muscle). Six to eight weeks of RNAi. Scale bars, 100 μm. Dorsal up. Numbers in each panel indicate number of animals displaying the result shown in the image out of total animals observed.

The online version of this article includes the following figure supplement(s) for figure 7:

**Figure supplement 1.** *map3k1* RNAi animals display ectopic teratoma-like growths.

were frequently isolated, as opposed to appearing exclusively in organized aggregates or in ectopically placed organs. Fifth, during regeneration or following eye resection in *map3k1* RNAi animals, progenitors could still be targeted to the correct TZ, indicating that the eye TZ remained present after *map3k1* RNAi. Finally, transplantation of EdU-labeled tissue grafts from *map3k1* RNAi animals into

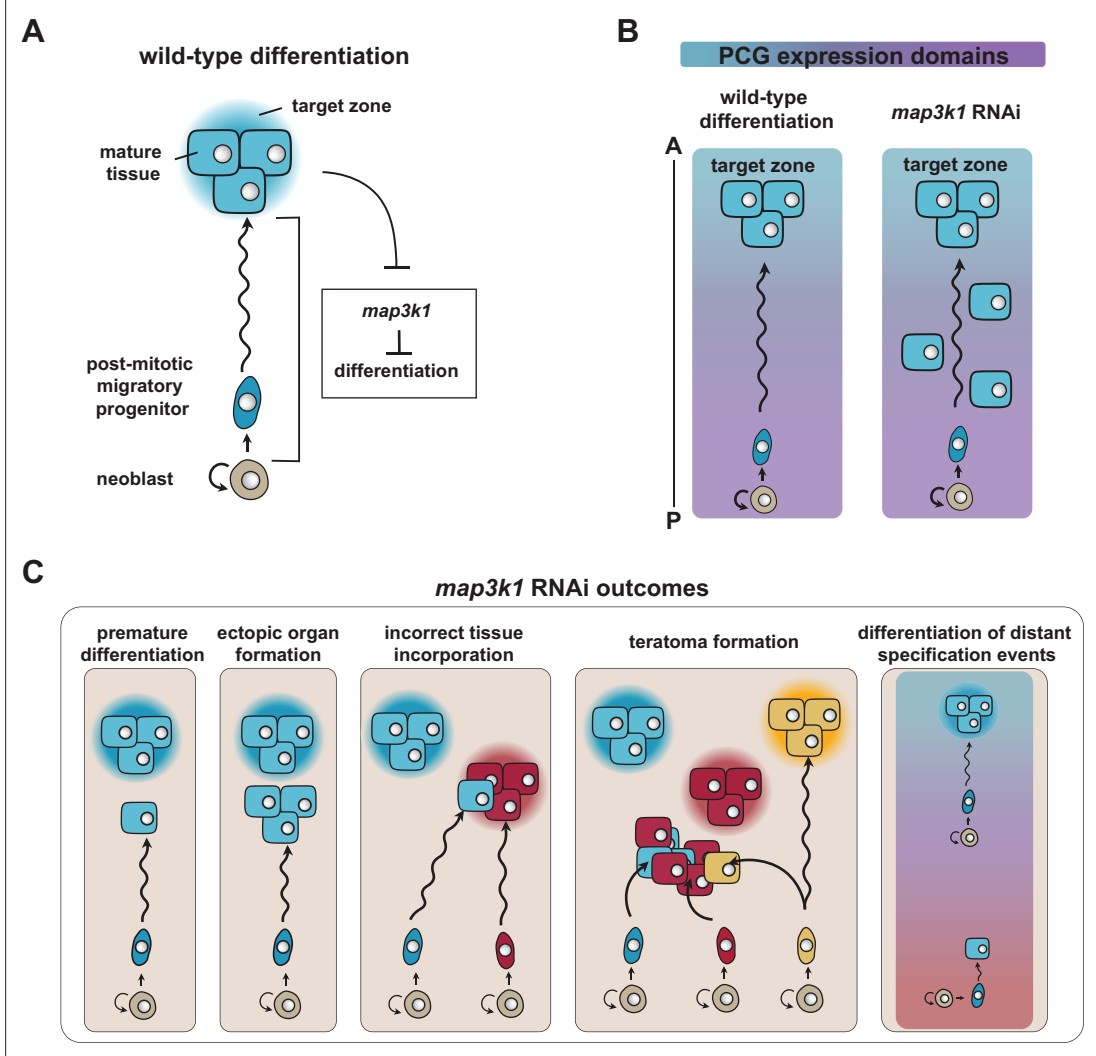

**Figure 8.** Model: Map3K1 restricts migratory progenitor differentiation until the correct target is reached. (**A**) Schematic showing the inhibition of differentiation of a migratory progenitor cell, via *map3k1*, until reaching its target tissue at the target zone. (**B**) Migratory precursors can differentiate in the incorrect position control gene (PCG) expression locations following *map3k1* RNAi. (**C**) Patterning abnormalities that can occur without suitable restriction of differentiation in migratory progenitors, demonstrating the patterning properties yielded by this mechanism.

the pre-pharyngeal region of control animals showed instances of ectopic differentiation in control host tissue.

Map kinase signaling cascades are highly conserved in the animal kingdom and are involved in diverse processes that regulate dynamic cellular behaviors via activation of Map kinases (e.g., ERK1/2, p38, JNK) that can act as transcriptional modifiers (*Widmann et al., 1999*; *Suddason and Gallagher, 2015*). Map kinase kinase kinases (MAP3Ks) are one of the first activated proteins in these cascades, often responding to receptor tyrosine kinase signaling at the cell membrane. In mammals, there are 24 characterized MAP3Ks: MAP3K1 through MAP3K21, B-Raf, C-Raf, and A Raf, which activate downstream MAP2K proteins through phosphorylation. Among the MAP3K proteins, MAP3K1 orthologs are the only MAP3K with a PHD domain, which enables a role in ubiquitination as well as kinase activity (*Pham et al., 2013*; *Suddason and Gallagher, 2015*). In mammals, MAP3K1 has been implicated in cell proliferation, differentiation, and cell death signaling (*Suddason and Gallagher, 2015*).

The molecular mechanism by which *map3k1* mediates its effect on planarian progenitors is still unclear. One scenario is that planarian post-mitotic progenitors are tuned to respond to a particular ECM or signaling environment (such as a PCG-related environment) to generate a molecular change that inactivates MAP3K1 signaling, such as by disengaging an RTK signal. Another possibility is that the progenitor migratory process itself could engage the MAP3K1 signal, enabling signal cessation with arrival at a target location. MAP3K1 can localize to focal adhesion complexes (*Christerson et al., 1999*; *Cuevas et al., 2003*), and has been implicated in integrin-mediated ECM-internalization mechanisms (*Martinez et al., 2024*) and interactions with Rho GTPases (*Zhang et al., 2005*; *Chen and Cobb, 2006*). In principle, one of these mechanisms could be connected to the regulation of planarian MAP3K1. When MAP3K1 is active, it could result in a transcriptional state that prevents full expression of differentiated factors required for maturation, tissue incorporation, and cessation of migration.

The defect in the spatial restriction of progenitor differentiation following *map3k1* RNAi can lead to dramatic tissue-patterning defects, including the differentiation of ectopic isolated cells (e.g., isolated PRNs), the emergence of ectopic organs, differentiated cells from one tissue type being present within an incorrect organ, and teratoma formation (*Figure 8C*). Furthermore, in the case of distant progenitor specification events, these cells can be prevented from ever differentiating with a *map3k1*-controlled mechanism, potentially accommodating noise in the spatial precision of stem cell fate specification by enabling the pruning of distant specified progenitors from the system. These tissue-patterning defects highlight the importance of spatially restricting the differentiation of post-mitotic progenitors for maintaining and regenerating adult pattern. Similar regulation might prove important during developmental contexts in many organisms involving cell migration and could be particularly important in adult regeneration where tissue scale can be large and adult progenitors, at least in some contexts, can be challenged to migrate large distances before differentiating.

Important mysteries remain regarding the specific ways in which *map3k1* regulates the patterning of different tissues. For example, ectopic dd_17258[+] neurons in *map3k1* RNAi animals were only out of place along the AP axis, and not on the ML axis; whereas ectopic eye cells were commonly out of place on both axes. This suggests that the ML component of differentiation regulation for dd_17258[+] neurons might not be *map3k1*-dependent. The differentiated patterns of some studied neural and gland cell populations were also unaffected in *map3k1* RNAi for unknown reasons, consistent with the possibility that *map3k1* is one of multiple mechanisms for regulating differentiation in pattern formation.

There is evidence for the role of various Map kinases (e.g., ERK, MEK, RAS, and p38) in planarian regeneration, particularly in blastema formation and wound response programs (*Tasaki et al., 2011a*; *Tasaki et al., 2011b*; *Owlarn et al., 2017*; *Wang et al., 2020*). It was suggested in another planarian species, *D. japonica,* that *map3k1* has a role in the scaling and patterning of the trunk and head regions of regenerating animals (*Hosoda et al., 2018*). Additionally, *map3k1* has been implicated in germ-cell proliferation and terminal differentiation of stem cells in the parasitic flatworm *E. multilocularis* through JNK signaling (*Stoll et al., 2021*). In (*Lo and Petersen, 2025*), gene function studies implicated both p38 and JNK Map kinases in the process regulated by *map3k1*. It will be of interest to further dissect the molecular role of *map3k1* in planarian progenitor differentiation and to determine whether *map3k1* orthologs have similar roles in regulating differentiation in other regenerative contexts. *map3k1* gene function is not well characterized in most invertebrate systems, including *Drosophila* and *C. elegans,* which have no identified *map3k1*

orthologs (*Widmann et al., 1999*). Planarians, therefore, present an attractive model for dissection of *map3k1* function.

Patterning systems in some organisms rely on spatially coarse and imperfect progenitor-specification systems, requiring the targeted migration of progenitors and local differentiation cues at target locations. This process involves progenitor transitions from spatially broad and disorganized to local and highly patterned structures (*Xiong et al., 2013*; *Park et al., 2023*). We suggest that in certain tissue-formation processes from dispersed progenitors, cells will be prevented from differentiation until suitable extrinsic cues have been detected or cellular interactions have occurred, and that this regulation will be fundamental to pattern formation. We suggest that *map3k1* acts within planarian progenitors to mediate such spatial restriction on differentiation, and that this is critical for preventing mistargeting of differentiation to incorrect locations and to prevent teratoma formation.

# Materials and methods

**Key resources table**

| Reagent type (species) or resource | Designation | Source or reference | Identifiers | Additional information |
|---|---|---|---|---|
| Gene (*S. mediterranea*) | *map3k1* | planmine database | dd_Smed_v6_5198_0_1 | https://planmine.mpinat.mpg.de/planmine/begin.do |
| Gene (*S. mediterranea*) | *opsin* | planmine database | dd_Smed_v6_15036_0_1 | https://planmine.mpinat.mpg.de/planmine/begin.do |
| Gene (*S. mediterranea*) | *catalase1* | planmine database | dd_Smed_v6_20433_0_1; dd_Smed_v6_32853_0_1 | https://planmine.mpinat.mpg.de/planmine/begin.do |
| Gene (*S. mediterranea*) | *glut3* | planmine database | dd_Smed_v6_79867_0_1 | https://planmine.mpinat.mpg.de/planmine/begin.do |
| Gene (*S. mediterranea*) | *tyrosinase* | planmine database | dd_Smed_v6_34399_0_1 | https://planmine.mpinat.mpg.de/planmine/begin.do |
| Gene (*S. mediterranea*) | *ovo* | planmine database | dd_Smed_v6_48430_0_1, dd_Smed_v6_10673_0_1 | https://planmine.mpinat.mpg.de/planmine/begin.do |
| Gene (*S. mediterranea*) | *gluR* | planmine database | dd_Smed_v6_16476_0_1 | https://planmine.mpinat.mpg.de/planmine/begin.do |
| Gene (*S. mediterranea*) | *dd_17258* | planmine database | dd_Smed_v6_17258_0_1 | https://planmine.mpinat.mpg.de/planmine/begin.do |
| Gene (*S. mediterranea*) | *NB.22.1e* | planmine database | dd_Smed_v6_680_0_1 | https://planmine.mpinat.mpg.de/planmine/begin.do |
| Gene (*S. mediterranea*) | *vitrin* | planmine database | dd_Smed_v6_1071_0_1 | https://planmine.mpinat.mpg.de/planmine/begin.do |
| Gene (*S. mediterranea*) | *mhc-1* | planmine database | dd_Smed_v6_249_0_1 | https://planmine.mpinat.mpg.de/planmine/begin.do |
| Gene (*S. mediterranea*) | *foxA* | planmine database | dd_Smed_v6_10718_0_1; clone Smed_02872_V2 | https://planmine.mpinat.mpg.de/planmine/begin.do |
| Gene (*S. mediterranea*) | *smedwi-1* | planmine database | dd_Smed_v6_659_0_1 | https://planmine.mpinat.mpg.de/planmine/begin.do |

*Continued on next page*

*Continued*

| Reagent type (species) or resource | Designation | Source or reference | Identifiers | Additional information |
|---|---|---|---|---|
| Gene (*S. mediterranea*) | dd_8476 | planmine database | dd_Smed_v6_8476_0_1 | https://planmine.mpinat.mpg.de/planmine/begin.do |
| Gene (*S. mediterranea*) | dd_7131 | planmine database | dd_Smed_v6_7131_0_1 | https://planmine.mpinat.mpg.de/planmine/begin.do |
| Gene (*S. mediterranea*) | dd_9223 | planmine database | dd_Smed_v6_9223_0_1 | https://planmine.mpinat.mpg.de/planmine/begin.do |
| Gene (*S. mediterranea*) | gad | planmine database | dd_Smed_v6_12653_0_1 | https://planmine.mpinat.mpg.de/planmine/begin.do |
| Gene (*S. mediterranea*) | slit | planmine database | dd_Smed_v6_12111_0_1 | https://planmine.mpinat.mpg.de/planmine/begin.do |
| Gene (*S. mediterranea*) | cintillo | planmine database | dd_Smes_g4_102 | https://planmine.mpinat.mpg.de/planmine/begin.do |
| Gene (*S. mediterranea*) | notum | planmine database | dd_Smed_v4_24180_0_1 | https://planmine.mpinat.mpg.de/planmine/begin.do |
| Gene (*S. mediterranea*) | chat | planmine database | dd_Smed_v6_6208_0_1 | https://planmine.mpinat.mpg.de/planmine/begin.do |
| Gene (*S. mediterranea*) | pc2 | planmine database | dd_Smed_v6_1566_0_1 | https://planmine.mpinat.mpg.de/planmine/begin.do |
| Gene (*S. mediterranea*) | ndl-2 | planmine database | dd_Smed_v4_8340_0_1 | https://planmine.mpinat.mpg.de/planmine/begin.do |
| Gene (*S. mediterranea*) | ndl-3 | planmine database | dd_Smed_v4_6604_0_1 | https://planmine.mpinat.mpg.de/planmine/begin.do |
| Gene (*S. mediterranea*) | ndl-5 | planmine database | dd_Smed_v4_5102_0_1 | https://planmine.mpinat.mpg.de/planmine/begin.do |
| Gene (*S. mediterranea*) | sfrp1 | planmine database | dd_Smed_v4_13985_0_1 | https://planmine.mpinat.mpg.de/planmine/begin.do |
| Gene (*S. mediterranea*) | wnt11-1 | planmine database | dd_Smed_v4_14391_0_1 | https://planmine.mpinat.mpg.de/planmine/begin.do |
| Gene (*S. mediterranea*) | wnt-1 | planmine database | dd_Smed_v4_28398_0_1 | https://planmine.mpinat.mpg.de/planmine/begin.do |
| Gene (*S. mediterranea*) | wntP-2 | planmine database | dd_Smed_v4_7326_0_1 | https://planmine.mpinat.mpg.de/planmine/begin.do |
| Gene (*S. mediterranea*) | ptk7 | planmine database | dd_Smed_v4_6999_0_1 | https://planmine.mpinat.mpg.de/planmine/begin.do |

*Continued on next page*

*Continued*

| Reagent type (species) or resource | Designation | Source or reference | Identifiers | Additional information |
|---|---|---|---|---|
| Gene (*S. mediterranea*) | *sp5* | planmine database | dd_Smed_v4_7824_0_1 | https://planmine.mpinat.mpg.de/planmine/begin.do |
| Gene (*S. mediterranea*) | *axin-B* | planmine database | dd_Smed_v4_5531_0_1 | https://planmine.mpinat.mpg.de/planmine/begin.do |
| Gene (*S. mediterranea*) | *prep* | planmine database | dd_Smed_v4_8606_0_1 | https://planmine.mpinat.mpg.de/planmine/begin.do |
| Gene (*S. mediterranea*) | *colF2* | planmine database | dd_Smed_v6_702_0_1 | https://planmine.mpinat.mpg.de/planmine/begin.do |
| Gene (*S. mediterranea*) | *mag1 (H.1.3b)* | planmine database | dd_Smed_v6_769_0_1 | https://planmine.mpinat.mpg.de/planmine/begin.do |
| Gene (*S. mediterranea*) | *estrella* | planmine database | dd_Smed_v6_1792_0_1 | https://planmine.mpinat.mpg.de/planmine/begin.do |
| Gene (*C. elegans*) | *unc-22* | WormBase | WBGene00006759 | |
| Strain, strain background | *Escherichia coli DH5α – CGSC strain* | *E. coli* Genetic Stock Center (CGSC), Yale University | CGSC #7750; RRID:SCR_002950 | Obtained in commercial kit Cat # NEBC2987H |
| Strain, strain background | Asexual *S. mediterranea* strain CIW4 | Laboratory of Alejandro Sánchez Alvarado, Stowers Institute | RRID:NCBITaxon:79327 | Clonal strain propagated in this lab from single animal |
| Antibody | TSA Plus DNP (HRP) System (signal amplification/detection kit) (Sheep polyclonal) | Akoya Biosciences | Cat # NEL747A001KT | 1:100 |
| Antibody | Anti-Fluorescein-POD, Fab fragments (Sheep polyclonal) | Roche (11426346910) | RRID:AB_840257 | 1:1500 |
| Antibody | anti-DIG-POD (Sheep polyclonal) | Roche (11207733910) | RRID:AB_514500 | 1:2000 |
| Antibody | Arrestin (VC-1) (Mouse monoclonal) | From the lab of Kiyokazu Agata | | 1:7500 |
| Sequence-based reagent | PWR.AA2 | GGGCGAATTGGGTACCGGG | | 5′ primer (5′–3′) |
| Sequence-based reagent | CP.D.47 | GAAGTAATACG ACTCACTATAGGGA GAAAGCTGGAG CTCCACCGCGG | | 3′ primer with T7 promotor region (5′–3′) |
| Sequence-based reagent | CP.C.21 | GAAGTAATACGACTCACT ATAGGGAGAGGG CGA ATT GGGTACCGGG | | 5′ primer with T7 promotor (5′–3′) |
| Sequence-based reagent | CP.C.22 | AAGCTGGAGCTCCACCGCGG | | 3′ primer (5′–3′) |
| Sequence-based reagent | DNP-11-UTP | PerkinElmer | Cat # NEL555001EA | |

*Continued on next page*

*Continued*

| Reagent type (species) or resource | Designation | Source or reference | Identifiers | Additional information |
|---|---|---|---|---|
| Sequence-based reagent | DIG RNA Labeling Mix (10X) | Roche/Sigma-Aldrich | Cat # 11277073910 | |
| Sequence-based reagent | Fluorescein RNA Labeling Mix | Roche/Sigma-Aldrich | Cat # 11685619910 | |
| Sequence-based reagent | pGEM-T Easy backbone | Promega | RRID:Addgene_122563 | |
| Commercial assay or kit | Superscript III | Invitrogen | Cat # 12574026 | Generating cDNA library |
| Commercial assay or kit | NEB 5-alpha Competent *E. coli* (High Efficiency) | New England Biolabs | Cat # NEBC2987H | |
| Commercial assay or kit | pGEM-T Easy Vector Systems | Promega | Cat # A1360/A1380 | |
| Commercial assay or kit | Riboprobe System Components and Buffers | Promega | Cat # P1121 | |
| Commercial assay or kit | T7 RNA polymerase | Promega | Cat # P2075 | |
| Chemical compound, drug | F-ara-EdU/ 2'-Deoxy- 2'-fluoro-5- ethynyluridine | Click Chemistry Tools | Cat # SKU: CCT-1403-500 | |
| Chemical compound, drug | TAMRA-Azide- fluor 545 | Sigma-Aldrich | Cat # SKU: 760757-1MG | |
| Software, algorithm | GraphPad prism 9 | GraphPad Software, San Diego, CA, USA | RRID:SCR_002798 | |
| Software, algorithm | R studio | RStudio, PBC, Boston, MA, USA | RRID:SCR_000432 | |

## Animal husbandry and surgery

Asexual *S. mediterranea* clonal strain CIW4 was used for all experiments. Animals were cultured in static 1x Montjuic water (1.6 mmol/l NaCl, 1.0 mmol/l CaCl$_2$, 1.0 mmol/l MgSO$_4$, 0.1 mmol/l MgCl$_2$, 0.1 mmol/l KCl, and 1.2 mmol/l NaHCO$_3$ prepared in Milli-Q water) at 20°C. Amputations were performed under cold conditions (~4°C) with a scalpel. Animals were fed homogenized beef liver weekly, with water changed biweekly. Animals were starved for approximately 7 days before experiments.

## Molecular cloning

A cDNA library was generated with RNA isolated from whole animals. All genes were amplified with gene-specific primers containing adaptors for 5' and 3' primer regions. Amplicons were ligated into a pGEM-T Easy backbone using the pGEM-T Easy Vector Systems kit (Promega). Plasmids were transformed into NEB 5-alpha Competent *E. coli* (High Efficiency; New England Biolabs) and miniprepped.

## In vitro transcription of RNA probes and dsRNA

Riboprobe System Components and Buffers (Promega) were used for in vitro transcription of dsRNA and RNA probes using T7 polymerase. RNA probes were transcribed using DIG, FITC, or DNP-modified nucleotides, allowing for signal amplification with conjugated DIG, FITC, and DNP antibodies. dsRNA was resuspended in water and RNA probes were resuspended in deionized formamide.

## Whole-mount FISH

Animal mucus was removed using 5% *N*-acetylcysteine in PBS; animals were then fixed with 4% formaldehyde in PBST for 20 min, with rocking. Animals were then washed with PBST, incubated in 1:1 PBST: methanol, then stored in 100% methanol at −20°C until ready for bleaching. Animals were moved into mesh baskets in a 24-well plate where all remaining steps were carried out. Animals were placed on a light source to bleach for 1.5 hr in a bleaching solution (5% formamide, 0.5x SSC, and 1.2% hydrogen peroxide). After two PBST washes, animals were then treated with 5 mg/ml Proteinase K for 10 min, followed by 4% formaldehyde post-fixation in PBST.

Probes were diluted in Hybe solution (1:800) (50% deionized formamide, 5x SSC, 1 mg/ml yeast RNA, 1% Tween-20, 5% dextran sulfate), and left to incubate overnight. The following days, we performed antibody incubations at 4°C overnight using anti-DIG-POD 1:1500, Roche; 10% western blocking solution (Roche) anti-FITC-POD (1:2000, Roche; 5% horse serum, 5% western blocking solution), and anti-DNP-HRP (1:100, PerkinElmer; blocking solution with 10% inactivated Horse Serum). Tyramide signal amplification involved incubating in rhodamine (1:1000), fluorescein (1:1500), or Cy5 (1:300) in borate buffer (0.1 M boric acid, 2 M NaCl, pH 8.5) containing 0.0003% hydrogen peroxide for 10 min. Samples were incubated in 1% sodium azide for 2 hr to inactivate the HRP. Blocking and antibody incubations then occurred for detection of the second probe. Animals were incubated overnight in 1 mg/ml DAPI solution at 4°C. Animals were mounted on coverslips in ProLong Gold Antifade Mountant (Thermo Fisher).

## EdU labeling and detection

F-ara-EdU (Click Chemistry Tools) was diluted in Dimethyl sulfoxide (DMSO) to 200 mg/ml, then diluted in static 1x Montjuic water to 1.25 mg/ml. Animals were split into 10 animals per well in a 12-well plate, then soaked in 1.25 mg/ml EdU solution for 20 hr following 1 week of starvation. EdU solution was replaced with 5 mg/ml Instant Ocean Sea salt dissolved in Milli-Q water. Prior to probe hybridization in the in situ hybridization protocol, following proteinase K and 4% formaldehyde incubations, cells were incubated in a 'click reaction' – 1% 100 mM CuSO$_4$, 0.1% 10 mM TAMRA-Azidefluor 545 (Sigma-Aldrich), and 20% 50 mM ascorbic acid in PBS for 30 min in the dark, proceeded by six PBST washes and continuation of the probe hybridization step.

## RNA interference

*C. elegans unc-22* dsRNA was used as the negative control for all RNAi experiments. 50 µl of homogenized beef liver was mixed with 25 µl of dsRNA prep and 3 µl of a 1:1 mixture of MilliQ water and red food coloring. *map3k1* RNAi experiment durations ranged from 1 to 16 weeks of dsRNA feedings for RNAi. Time courses were conducted using animals fed 1, 2, and 3–4 weeks of dsRNA. Animals used to study outgrowth phenotypes were fixed between 8 and 12 weeks of RNAi, based on teratoma severity. Animals fed dsRNA for 3 and 4 weeks showed similar phenotype severity and were often analyzed as one group. Animals were given food for 1 hr, twice a week, for the first 8 feedings (4 weeks); animals were then given food once a week, for 30 min, for all subsequent feedings to prevent rapid growth and fissioning (separation of the tail from the body).

## EdU transplantation assays

EdU plug transplants were performed using *map3k1* or control RNAi animals, 12 hr following a 20-hr EdU pulse, as the donor to a recipient wild-type animal. Donor animals were anesthetized with 0.2% chlorotone solution, followed by an incubation in Holfreter's solution, then placed on an ice block covered in Whatman filter paper moistened with 1x Montjuic water to surgically manipulate with a clean scalpel. EdU-positive *map3k1* RNAi donor animals and wild-type recipients both had a center portion of their pre-pharyngeal regions removed. The pre-pharyngeal donor graft from the *map3k1* RNAi animal was placed in the EdU-negative wild-type recipient's pre-pharyngeal region. Recipients were then gently covered with cigarette paper soaked in chilled Holtfreter's solution and transferred to a small Petri dish with enough Holtfreter's solution to cover the bottom of the dish. Petri dishes were put at 10°C for 20 hr; the following day, transplant recipients were gently recovered and put into 1x Montjuic water containing 0.1% gentamicin (Gibco) to heal. Water was changed every 2 days, and animals were fixed at day 12 post-transplant.

## Shielded irradiation

Animals were irradiated using an X-Rad320, Precision X-Ray Irradiation chamber. For shielded irradiation experiments, animals were anesthetized with 0.2% chlorotone, then arranged on Whatman filter paper in a Petri dish sitting on ice. Animals were oriented to have their anterior half covered by the lead shield placed over the Petri dish. Samples were placed in the irradiation chamber and exposed to 3000 Rad of unidirectional X-irradiation. Animals were rescued with 1x Montjuic water and stored in 1x Montjuic water with 0.1% gentamicin (Gibco) to recover. Water was changed every 2 days. RNAi experiments were carried out starting 2 days after irradiation exposure. Anterior half-shielded animals were fed dsRNA for a period of 12–14 days before fixation, and posterior half-shielded animals were fed dsRNA for a period of 10–12 days before fixation; these ranges were dependent on the health of the animals at the time of fixation. If animals started showing slight signs of health decline (e.g., small lesions, slight head regression, bloating), they were fixed on that day.

## Regeneration assays

All tissue resections and amputations were performed by placing animals on wet filter paper on top of a cold block to minimize movements during surgeries. Animals were kept moist throughout all procedures with 1x Montjuic water. Eyes were resected using a small scalpel. Curved edges were created by lightly tapping the tip of the scalpel on a clean, hard surface. The curved edge was used to poke and scoop out the eye in a poke and pull motion. Pharynges were resected by puncturing a diamond shape around the pharynx with a small scalpel, then gently removing the pharynx tissue. Animals were placed in 1x Montjuic water with 0.1% gentamicin (Gibco) to recover, and water was changed every 2 days.

## Image and statistical analyses

FISH images were analyzed using Fiji Software. The AP axis was binned into six regions according to anatomical landmarks: AP_1 (head tip → bottom of the brain), AP_2 (bottom of the brain → top of the pharynx), AP_3 (top of the pharynx → middle of the pharynx), AP_4 (middle of the pharynx → bottom of the pharynx), AP_5 (bottom of the pharynx → halfway between the bottom of the pharynx and the tail tip), and AP_6 (halfway between the bottom of the pharynx and the tail tip → tail tip). Each animal had one data point in each of the six AP bins. *map3k1* and control RNAi ectopic PRN and OC cell counts at 3–4 weeks were analyzed by generating a Poisson generalized linear mixed model, using AP bin as a random covariate. A Mann–Whitney $U$ test was used for dd_17258$^+$ and EdU$^+$ cell counts in transplants to account for non-normal data distributions. Any tail or head cell count data sets showing overdispersion – gland cells (dd_7131 and dd_8476) and *FoxA$^+$; smedwi-1$^-$* cell counts between the brain – were analyzed with a negative binomial regression, correcting for overdispersion. Two-tailed permutation tests, using 10,000 permutations per test, were carried out for *ovo$^+$* cell counts in the tail because of the low sample number and zero-inflated dataset. A binomial exact test was carried out on ectopic cells inside versus outside the *smedwi-1$^+$* zone after shielded irradiation and *map3k1* RNAi, assuming 50% probability of either outcome. Prism software was used to carry out Student's *t*-tests, Mann–Whitney $U$ tests, and binomial exact tests. R Studio was used to compute negative binomial regression, Poisson regression, and permutation tests.

# Acknowledgements

The authors thank members of the Reddien lab and Troy Whitfield for helpful comments and discussion. We acknowledge support from NIH R35 GM145345. PWR is an investigator of HHMI and an associate member of the Broad Institute. We thank the Eleanor Schwartz Charitable Foundation for support.

## Additional information

### Funding

| Funder | Grant reference number | Author |
|---|---|---|
| National Institutes of Health | R35 GM145345 | Peter W Reddien |
| Eleanor Schwartz Charitable Foundation | grant | Peter W Reddien |
| Howard Hughes Medical Institute | Investigator | Peter W Reddien |

The funders had no role in study design, data collection, and interpretation, or the decision to submit the work for publication.

### Author contributions

Bryanna Isela-Inez Canales, Conceptualization, Data curation, Formal analysis, Investigation, Methodology, Visualization, Writing – original draft, Writing – review and editing; Hunter O King, Conceptualization, Formal analysis, Investigation, Visualization, Writing – review and editing; Peter W Reddien, Conceptualization, Resources, Supervision, Funding acquisition, Investigation, Methodology, Writing – original draft, Project administration, Writing – review and editing

### Author ORCIDs

Bryanna Isela-Inez Canales ⑤ https://orcid.org/0000-0002-0693-3632
Hunter O King ⑤ https://orcid.org/0009-0001-9823-1856
Peter W Reddien ⑤ https://orcid.org/0000-0002-5569-333X

Reviewer #1 (Public review): https://doi.org/10.7554/eLife.106439.3.sa1
Reviewer #2 (Public review): https://doi.org/10.7554/eLife.106439.3.sa2
Author response https://doi.org/10.7554/eLife.106439.3.sa3

## Additional files

### Supplementary files

MDAR checklist

### Data availability

The data used in *Figure 5—figure supplement 2* was a 10 X scRNA-seq dataset generated in *King et al., 2024*. This dataset is available as PRJNA1067154 (SRA) '10 X scRNA-seq of Schmidtea: X1 Neoblasts and G0 Progenitor Cells'.

The following previously published dataset was used:

| Author(s) | Year | Dataset title | Dataset URL | Database and Identifier |
|---|---|---|---|---|
| King HO, Owusu-Boaitey KE, Fincher CT, Reddien PW | 2024 | 10X scRNA-seq of Schmidtea: X1 Neoblasts and G0 Progenitor Cells | https://www.ncbi.nlm.nih.gov/sra?term=PRJNA1067154 | NCBI Sequence Read Archive, PRJNA1067154 |

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
