## [Editor Report · eLife Assessment]

This **important** study examines the role of map3k1, a MAP3K family member that has both kinase and ubiquitin ligase domains, in the differentiation of progenitors in the flatworm Planaria. The **convincing** analyses demonstrate that map3k1 acts within progenitors to restrict their premature differentiation and to prevent formation of teratomas. This work would be of interest to researchers in the fields of regeneration, developmental biology, and aging.

---

## [Referee Report · Reviewer #1 (Public review)]

Summary:

The authors assess the role of map3k1 in adult Planaria through whole body RNAi for various periods of time. The authors' prior work has shown that neoblasts (stem cells that can regenerate the entire body) for various tissues are intermingled in the body. Neoblasts divide to produce progenitors that migrate within a "target zone" to the "differentiated target tissues" where they differentiate into a specific cell type. Here the authors show that map3k1-i animals have ectopic eyes that form along the "normal" migration path of eye progenitors, ectopic neurons and glands along the AP axis and pharynx in ectopic anterior positions. The rest of the study shows that positional information is largely unaffected by loss of map3k1. However, loss of map3k1 leads to premature differentiated of progenitors along their normal migratory route. They also show that "long-term" whole body depletion of map3k1 results in mis-specified organs and teratomas. In short, this study convincingly demonstrates that in planaria, map3k1 maintains progenitor cells in an undifferentiated state, preventing premature fate commitment until they encounter the appropriate signals, either positional cues within a designated region or contact-dependent inputs from surrounding tissues.

Strengths:

(1) The study has appropriate controls, sample sizes and statistics.

(2) The work is high-quality.

(3) The conclusions are supported by the data.

(4) Planaria is a good system to analyze the function of map3k1, which exists in mammals but not other invertebrates.

Weaknesses:

None noted.

---

## [Referee Report · Reviewer #2 (Public review)]

Summary:

The flatworm planarian *Schmidtea mediterranea* is an excellent model for understanding cell fate specification during tissue regeneration and adult tissue maintenance. Planarian stem cells, known as neoblasts, are continuously deployed to support cellular turnover and repair tissues damaged or lost due to injury. This reparative process requires great precision to recognize the location, timing, and cellular fate of a defined number of neoblast progeny. Understanding the molecular mechanisms driving this process could have important implications for regenerative medicine and enhance our understanding of how form and function are maintained in long-lived organisms such as humans. Unfortunately, the molecular basis guiding cell fate and differentiation remains poorly understood.

In this manuscript, Canales et al. identified the role of the map3k1 gene in mediating the differentiation of progenitor cells at the proper target tissue. The map3k1 function in planarians appears evolutionarily conserved as it has been implicated in regulating cell proliferation, differentiation, and cell death in mammals. The results show that the downregulation of map3k1 with RNAi leads to spatial patterning defects in different tissue types, including the eye, pharynx, and the nervous system. Intriguingly, long-term map3k1-RNAi resulted in ectopic outgrowths consistent with teratomas in planarians. The findings suggest that map3k1 mediates signaling, regulating the timing and location of cellular progenitors to maintain correct patterning during adult tissue maintenance.

Strengths:

The authors provide an entry point to understanding molecular mechanisms regulating progenitor cell differentiation and patterning during adult tissue maintenance.

The diverse set of approaches and methods applied to characterize map3k1 function strengthens the case for conserved evolutionary mechanisms in a selected number of tissue types. The creativity using transplantation experiments is commendable, and the findings with the teratoma phenotype are intriguing and worth characterizing.

Weaknesses:

The authors have satisfactorily addressed our previous concerns.

---

## [Author Response]

The following is the authors’ response to the original reviews.

**Public Reviews:**

**Reviewer #1 (Public review):**
Summary:The authors assess the role of map3k1 in adult Planaria through whole body RNAi for various periods of time. The authors' prior work has shown that neoblasts (stem cells that can regenerate the entire body) for various tissues are intermingled in the body. Neoblasts divide to produce progenitors that migrate within a "target zone" to the "differentiated target tissues" where they differentiate into a specific cell type. Here the authors show that map3k1-i animals have ectopic eyes that form along the "normal" migration path of eye progenitors (Fig. 1), ectopic neurons and glands along the AP axis (Fig. 2) and pharynx in ectopic anterior positions (Fig. 3). The rest of the study show that positional information is largely unaffected by loss of map3k1 (Fig. 4,5). However, loss of map3k1 leads to premature differentiated of progenitors along their normal migratory route (Fig. 6). They also show that an ill-defined "long-term" whole body depletion of map3k1 results in mis-specified organs and teratomas.Strengths:(1) The study has appropriate controls, sample sizes and statistics.(2) The work appears to be high-quality.(3) The conclusions are supported by the data.(4) Planaria is a good system to analyze the function of map3k1, which exists in mammals but not in other invertebrates.Weaknesses:(1) The paper is largely descriptive with no mechanistic insights.

The mechanistic insights we aim to address are primarily at the cellular systems level – how adult progenitor cells produce pattern. Specifically, we uncovered strong evidence that regulation of differentiation is an active process occurring in migratory progenitors and that this regulation is a major component of pattern formation during the adult processes of tissue turnover and regeneration. The map3k1 phenotype provided a tool used to reveal the existence of this regulation, and to understand the patterning abnormalities prevented by this regulatory mechanism. We updated the text in several places to make clearer some of this emphasis. For example, in the Discussion: "We suggest that differentiation is restricted during migratory targeting as an essential component of pattern formation, with the map3k1 RNAi phenotype indicating the existence and purpose of this element of patterning."

Naturally, identifying a particular molecule involved in this process is of interest for understanding molecular mechanism; this would allow for comparison to other cellular systems in other organisms and would focus future molecular inquiry. Future molecular studies into the mechanism of Map3k1 regulation and its downstream signaling will be fascinating as next steps towards understanding the process at the molecular level more deeply. We also added some discussion considering the types of upstream activation cues that could potentially be associated with Map3k1 regulation to suppress differentiation.

(2) Given the severe phenotypes of long-term depletion of map3k1, it is important that this exact timepoint is provided in the methods, figures, figure legends and results.

We removed the use of the term “long-term” and instead added timepoints used to all figure legends. We also added a summary of timepoints used in the methods section and included RNAi timepoint labels in figures where a phenotype progression over time is relevant to interpretation. For timecourses, we also added suitable time information to text in the results.

(3) Figure 1C, the ectopic eyes are difficult to see, please add arrows.

To improve visualization, we replaced the example animal in the original Figure 1C with one that has a stronger phenotype, including arrows pointing to every ectopic event. Additionally, we included magnified images of optic cup cells and photoreceptor neurons in the trunk and tail region. This is now Figure 1B.

(4) line 217 - why does the n=2/12 animals not match the values in Figure 3B, which is 11/12 and 12/12. The numbers don't add up. Please correct/explain.

In Figure 3B in the submitted version (3/18 had cells in the tail) had more animals scored (6 animals from a replicate experiment where 1/6 showed the cells in the tail) than the total scored (2/12 had cells in the tail) in the text, which did not have the animals from the replicate added during writing. The results are the same, just different sample sizes were noted in those locations and we fixed this issue. In the updated Figure 3, the order of presentation has shifted (e.g., prior 3B is now in 3C and Figure 3_figure supplement 1). We made sure to include numbers to all figure panels.

(5) Figure panels do not match what is written in the results section. There is no Figure 6E. Please correct.

Thank you for catching this. We have gone through figures and text after editing to make sure that text callouts are appropriately matched to the figures.

**Reviewer #2 (Public review):**
Summary:The flatworm planarian *Schmidtea mediterranea* is an excellent model for understanding cell fate specification during tissue regeneration and adult tissue maintenance. Planarian stem cells, known as neoblasts, are continuously deployed to support cellular turnover and repair tissues damaged or lost due to injury. This reparative process requires great precision to recognize the location, timing, and cellular fate of a defined number of neoblast progeny. Understanding the molecular mechanisms driving this process could have important implications for regenerative medicine and enhance our understanding of how form and function are maintained in long-lived organisms such as humans. Unfortunately, the molecular basis guiding cell fate and differentiation remains poorly understood.In this manuscript, Canales et al. identified the role of the map3k1 gene in mediating the differentiation of progenitor cells at the proper target tissue. The map3k1 function in planarians appears evolutionarily conserved as it has been implicated in regulating cell proliferation, differentiation, and cell death in mammals. The results show that the downregulation of map3k1 with RNAi leads to spatial patterning defects in different tissue types, including the eye, pharynx, and the nervous system. Intriguingly, long-term map3k1-RNAi resulted in ectopic outgrowths consistent with teratomas in planarians. The findings suggest that map3k1 mediates signaling, regulating the timing and location of cellular progenitors to maintain correct patterning during adult tissue maintenance.Strengths:The authors provide an entry point to understanding molecular mechanisms regulating progenitor cell differentiation and patterning during adult tissue maintenance.The diverse set of approaches and methods applied to characterize map3k1 function strengthens the case for conserved evolutionary mechanisms in a selected number of tissue types. The creativity using transplantation experiments is commendable, and the findings with the teratoma phenotype are intriguing and worth characterizing.

Thank you to the reviewer for the positive feedback

Weaknesses:The article presents a provocative idea related to the importance of positional control for organs and cells, which is at least in part regulated by map3k1. Nonetheless, the role of map3k1 or its potential interaction with regulators of the anterior-posterior, mediolateral axes, and PCGs is somewhat superficial. The authors could elaborate or even speculate more in the discussion section and the different scenarios incorporating these axial modulators into the map3k1 model presented in Figure 8

First, to strengthen the support for our finding that positional information is largely unaffected in map3k1 RNAi animals, we added data regarding the expression of additional relevant position control genes (PCGs) –ndl-4, ptk7, sp5, and wnt11-1 – to the PCG panel in Figure 5. The expression domain of ndl-4, an FGF receptor-like protein family member that contributes to head patterning and anterior pole maintenance, was normal in map3k1 RNAi. wnt11-1, a PCG with expression concentrated in the posterior end of the animal and with expression dependent on general Wnt activity, was also normal in map3k1 RNAi animals. ptk7, RNAi of which can result in supernumerary pharynges, also showed normal expression in map3k1 RNAi animals. Finally, sp5, a Wnt-activated gene with expression in the tail, also showed normal expression in map3k1 RNAi animals.

Second, to further support the conclusion that cells are not suitably responding to positional information after map3k1 RNAi, which we argue normally dictates where differentiation should occur, we added examples of differentiated cell types that are ectopically positioned within an atypical PCG expression domain for that cell type (Figure 5C). This underscores that following map3k1 RNAi the PCG expression domains do not change, but the pattern of differentiated cell types relative to these domains does shift. We also added data showing that regenerating tails had a proper wntP-2 gradient, but an anterior regenerating pharynx appeared outside of this wntP-2^+^ zone and inside of an ndl-5^+^ zone (Figure 5- figure supplement 1E). We added some discussion of these new data in the Figure 5 results section. We also noted, regarding independent recent map3k1 work (Lo, 2025), some evidence exists that a minor posterior shift in ndl-5 expression can occur after map3k1 RNAi.

Next, we added a new element to the model figure (Figure 8B) depicting that PCG expression domains remain normal after map3k1 RNAi, with ectopic differentiation occurring in an incorrect positional information environment. We refer to this new panel in the discussion: "We suggest that map3k1 is not required for the spatial distribution of progenitor-extrinsic differentiation-promoting cues themselves, but for progenitors to be restricted from differentiating until these cues are received (Figure 8B)."; we then follow this statement with a summary in the Discussion of six pieces of evidence that support this model.

Finally, we added some additional text to the discussion section about candidate mechanisms by which extrinsic cues could potentially regulate Map3k1, pointing to potential future inquiry directions. We suggest that map3k1 is not involved in regulating PCG activity domains themselves, but instead acts as a brake on differentiation within migratory progenitors through active signaling. This brake is then lifted when the progenitors hit their correct PCG-based migratory target, or when they hit their target tissue. How that occurs mechanistically is unknown. One scenario is that each progenitor is tuned to respond to a particular PCG-regulated environment (such as a particular ECM or signaling environment) to generate a molecular change that inactivates Map3K1 signaling, such as by inactivating or disengaging an RTK signal. Alternatively, the migratory process in progenitors could engage the Map3K1 signal, enabling signal cessation with arrival at a target location. When Map3K1 is active it could result in a transcriptional state that prevents full expression of differentiated factors required for maturation, tissue incorporation, and cessation of migration. These considerations are now added to the discussion.

The article can be improved by addressing inconsistencies and adding details to the results, including the main figures and supplements. This represents one of the most significant weaknesses of this otherwise intriguing manuscript. Below are some examples of a few figures, but the authors are expected to pay close attention to the remaining figures in the paper.Details associated with the number of animals per experiment, statistical methods used, and detailed descriptions of figures appear inconsistent or lacking in almost all figures. In some instances, the percentage of animals affected by the phenotype is shown without detailing the number of animals in the experiment or the number of repeats. Figures and their legends throughout the paper lack details on what is represented and sometimes are mislabeled or unrelated.

We endeavored to ensure that these noted details are present throughout the legends and figures for all figure panels.

Specifically, the arrows in Figure 1A are different colors. Still, no reasoning is given for this, and in the exact figure, the top side (1A) shows the percentages and the number of animals below.

The only reason for the different colored arrows was for visibility purposes. To avoid confusion, we now use white arrows for all FISH images in figure 1, and where ever else possible. We also replaced the percentages with n numbers in the bottom left corner of the live images in Figure 1A.

Conversely, in Figures 1B, C, and D, no details on the number of animals or percentages are shown, nor an explanation of why opsin was used in Figure 1A but not 1B.

The original Figure 1B represented a few different examples of ectopic eye/eye cell patterns in the map3k1 RNAi animals to demonstrate the variable and disorganized nature of the phenotype. To address this, we added further explanation in the legend. We also merged 1A and 1B for simplicity of interpretation. opsin was used in Figure 1A to label cell bodies of photoreceptors. anti-Arrestin was used in the example FISH images to see if these cells were interconnected via projections, which we now clarify in the legend and in the text.

Is Figure 1B missing an image for the respective control? Figure 1C needs details regarding what the two smaller boxes underneath are.

The control for Figure 1B was in Figure 1A; the merger of Figures 1A/B should address this. Boxes in Figure 1C were labelled with numbers corresponding to the image above them.

Figure 1C could use an AP labeling map in 10 days (e.g., AP6 has one optic cup present). Figure 1C and F counts do not match.

We added a cartoon to 1C to show the region imaged. Note that the 36d trunk image (now Fig. 1B) has now been replaced with a full animal image and magnified boxes that show locations of example ectopic cells. That cell in 1C was categorized as in AP5. Note that additional animals were analyzed and data added to the distribution (now Fig. 1D).

In Figure 1C, we do not know the number of animals tested, controls used, the scale bar sizes in the first two images, nor the degree of magnification used despite the pharynx region appearing magnified in the second image. Figure 1C is also shown out of chronological order; 36 days post RNAi is shown before 10 days post RNAi. Moreover, the legends for Figures 1C and 1D are swapped.

We have endeavored to ensure sample numbers, control images, and appropriate scale bar notation in legends are present for all images. Figure 1C has now been split into two panels: Figure 1B and Figure 1C. It does not follow a chronological order in presentation for the following logic flow: we uncover and describe the phenotype; then, with knowledge of the defect, we walk back to see how early the phenotype starts after RNAi and what the pattern of ectopic cell distribution is when the phenotype starts to emerge (using the knowledge of which cells are affected from the overt phenotype described in 1A/B).

Additionally, Figure 1F and many other figures throughout the paper lack overall statistical considerations. Furthermore, Figure 1F has three components, but only one is labeled. Labeling each of them individually and describing them in the corresponding figure legend may be more appropriate.

The main point of the graphs in 1F (now 1D) was the overt overall pattern difference with the wild-type, which never has ectopic eye cells in the midbody or tail, and that the ectopic eye cells progress throughout the entire AP axis. However, we concur that a statistical test is a reasonable thing to show here and that is now included in the legend. The 3 components (in Figure 1F, now Figure 1D) where kept together with one figure label (D) for simplicity, but were rearranged (top and bottom) with a cartoon to the side and with modified labeling for extra clarity.

Figure 2C shows images of gene expression for two genes, but the counts are shown for only one in Figure 2D. It is challenging to follow the author's conclusions without apparent reasoning and by only displaying quantitative considerations for one case but not the other. These inconsistencies are also observed in different figures.

In Figure 2C, FISH images of cintillo+ and dd_17258+ neurons are shown to display the specificity of this effect to some neurons and not others. Because cintillo+ cells did not expand at all (n=24/24 animals), the counts for them would all be zero values. We only counted data for dd_17258 cells because it was the neuron that expanded compared to the control animals. We have now added a note in the legend explaining this.

In Figure 2D, 24/24 animals were reported to show the phenotype, but only eight were counted (is there a reason for this?).

8 animals were used to quantitatively characterize the spread of cells along the AP axis, as it was deemed an adequate sample size to capture the change in distribution of 17258+ cells from being head restricted to being present throughout the body. Through multiple cohorts of animals in replicates, a total of 24/24 examined animals showed this expansion phenotype. Double FISH experiments were additionally carried out using dd_17258 and various PCGs; these data are now included in Figure 5C, and these animals were added to the total counts regarding quantitative analysis of the phenotype in Figure 2D.

In Figure 2E, the expression for three genes is shown, with some displaying anterior and posterior regions while others only show the anterior picture. Is there a particular reason for this?

The original first panel in Figure 2E showed an example of a non-expanding gland cell type, dd_9223, which is very restricted to the head in both control and map3k1 RNAi animals. Because we did not observe a phenotype for this cell type (no cells in all control and map3k1 RNAi animal tails), we only included tail images of cell types that showed an abnormal phenotype with clear expanded to the posterior (dd_8476 and dd_7131). However, we have now included tail images of dd_9223 cells and added data for dd_9223 to the graph in Figure 2E.

Also, in Figure 2F, the counts are shown for only the posterior region of two genes out of the three displayed in Figure 2E. It is unclear why the authors do not show counts for the anterior areas considered in Figure 2E. Furthermore, the legend for Figure 2D is missing, and the legend for 2F is mislabeled as a description for Figure 2D.

We now include tail images for dd_9223 in Figure 2E to show that there are no ectopic cells in tails. We did not originally include counts of dd_9223 because there was no phenotype observed. dd_7131 and dd_8476 cell types appeared in the posterior of even control animals at a low frequency, unlike dd_9223 cells. However, we did now add counts for dd_9223 tail regions in the graph. We did not count the anterior regions of the animal because our goal was to show data for the visible phenotype (ectopic cells in the tail) not only with an example image, but also by showing the number of cells in the tail with a graph and statistical test. Legends have been updated with correct details.

Supplement Figure 1 B reports data up to 6 weeks, but no text in the manuscript or supplement mentions any experiment going up to 6 weeks. There are no statistics for data in Supplement Figure 1E. Any significance between groups is unclear.

More details about the RNAi feeding schedules have been added in the methods section. All RNAi timepoints are now specified specifically in the legends. The Figure 1F and Figure 1- figure supplement 1E (additional data: ovo^+^; smedwi-1^-^ cell counts) and legends now mention the statistical tests performed and annotations (not significant *ns) or p values have been added to the graphs. For simplicity, we decided to include all smedwi-1+ counts together rather than splitting them into low and high smedwi-1+ cells, because we weren't really making any claims about low and high cells.

**Recommendations for the authors:**

**Reviewer #1 (Recommendations for the authors):**
It would be good to acknowledge in the discussion the recent paper from the Petersen lab on map3k1, published in PLoS Genet 2025, especially if the results differ between the two labs.

We added reference/discussion regarding the recent PLoS Genetics Lo, 2025 map3k1 paper at several suitable points in the manuscript.

**Reviewer #2 (Recommendations for the authors):**
Please pay close attention to the description of experimental details and the consistency throughout the paper. It seems like the reader has to assume or come across information that is not readily available from the text or the legends in the paper. This is an interesting paper with intriguing findings. However, the version presented here appears rushed or put together on the flight.

Thank you for your thorough feedback. We have endeavored to ensure all appropriate details are present in figures and/or figure legends.